# Seismic imaging in the eastern Scandinavian Caledonides: siting the 2.5 km deep COSC-2 borehole, central Sweden

**C. Juhlin, P. Hedin, D. G. Gee, H. Lorenz, T. Kalscheuer, and P. Yan**

Department of Earth Sciences, Uppsala University, Uppsala, Sweden

Received: 1 December 2015 – Accepted: 3 December 2015 – Published: 15 January 2016

Correspondence to: P. Hedin (peter.hedin@geo.uu.se)

Published by Copernicus Publications on behalf of the European Geosciences Union.

**SED**

doi:10.5194/se-2015-129

Seismic imaging in the eastern Scandinavian Caledonides

C. Juhlin et al.

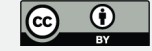

Discussion Paper | Discussion Paper | Discussion Paper | Discussion Paper | Discussion Paper |

**SED**

doi:10.5194/se-2015-129

**Seismic imaging in the eastern Scandinavian Caledonides**

C. Juhlin et al.

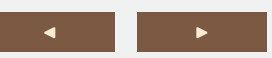

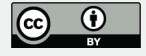

## Abstract

The Collisional Orogeny in the Scandinavian Caledonides (COSC) project, a contribution to the International Continental Scientific Drilling Program (ICDP), aims to provide a deeper understanding of mountain belt dynamics. Scientific investigations include a range of topics, from subduction-related tectonics to the present-day hydrological cycle. COSC investigations and drilling activities are focused in central Scandinavia where rocks from the mid to lower crust of the orogen are exposed near the Swedish-Norwegian border. Here, rock units of particular interest occur in the Seve Nappe Complex (SNC) of the so-called Middle Allochthon and include granulite facies migmatites (locally with evidence of ultra-high pressures) and amphibolite facies gneisses and mafic rocks. This complex overlies greenschist facies metasedimentary rocks of the dolerite-intruded Särv Nappes and underlying, lower grade Jämtlandian Nappes (Lower Allochthon). Reflection seismic profiles have been an important component in the activities to image the sub-surface structure in the area. Sub-horizontal reflections in the upper 1–2 km are underlain and interlayered with strong west- to northwest-dipping reflections, suggesting significant east-vergent thrusting. Two 2.5 km deep fully cored boreholes are a major component of the project which will improve our understanding of the sub-surface structure and tectonic history of the area. Borehole COSC-1, drilled in the summer of 2014, targeted the subduction-related Seve Nappe Complex and the contact with the underlying allochthon. The COSC-2 borehole will be located further east and investigate the lower grade, mainly Cambro-Silurian rocks of the Lower Allochthon, the main Jämtlandian décollement and penetrate into the crystalline basement rocks to identify the source of some of the northwest-dipping reflections. A series of high resolution seismic profiles have been acquired along a composite ca. 55 km long profile to help locate the COSC drillholes. We present here the results from this COSC-related composite seismic profile (CSP), including new interpretations based on previously unpublished data acquired between 2011 and 2014. These seismic data, along with shallow drillholes in the Caledonian thrust front and previously

acquired seismic, magnetotelluric, and magnetic data, are used to identify two potential drill sites for the COSC-2 borehole.

# 1 Introduction

Following the Ordovician closure of the Iapetus Ocean, major Caledonian orogeny involved continent collision and underthrusting of Baltica beneath Laurentia. Subduction-related metamorphism along the Baltica margin was taking place already in the early to middle Ordovician (Gee et al., 2012; Majka et al., 2012) and the initial stages of continent-continent collision are believed to have occurred around 445 Ma (e.g. Ladenberger et al., 2012, 2014). Thrust tectonics, which dominated throughout the collision, resulted in the emplacement of allochthonous units both westwards onto the Laurentian platform of Greenland (Higgins and Leslie, 2000) with displacements of at least 200 km, and eastwards onto the Baltoscandian platform with displacements of more than 400 km (Gee, 1978).

Towards the end of Caledonian Orogeny, in the early Devonian, the mountain belt was in many aspects comparable to the presently active Himalaya-Tibet Orogen (Dewey, 1969; Gee et al., 2010; Labrousse et al., 2010). Following post-orogenic collapse, extension and several hundred million years of erosion, the surface of the present day Caledonides cuts through the internal architecture of the paleo-orogen, revealing the nappe structure at mid-crustal depths. The Scandinavian mountains, the Scandes, have long been recognized as an excellent environment to study thrust tectonics (Törnebohm, 1888) and the processes involved in continent-continent collision (Gee, 1975; Hossack and Cooper, 1986).

Investigations of the Scandinavian Caledonides were intensified in the 1970's (Gee and Sturt, 1985) and our understanding has since then improved through continued geological (e.g. the many contributions in Corfu et al., 2014) and numerous geophysical (e.g. Dyrelius, 1980, 1986; Elming, 1988; Hurich et al., 1989; Palm et al., 1991; Hurich, 1996; Juhojuntti et al., 2001; Pascal et al., 2007; Korja et al., 2008; England and

Discussion Paper | Discussion Paper | Discussion Paper | Discussion Paper | Discussion Paper |

**SED**

doi:10.5194/se-2015-129

**Seismic imaging in the eastern Scandinavian Caledonides**

C. Juhlin et al.

Ebbing, 2012) studies. One key area of investigation (Dyrelius et al., 1980) has been along a profile crossing the mountain belt through the provinces of Jämtland (Sweden) and Tröndelag (Norway). Reflection seismic surveys were conducted along the Central Caledonian Transect (CCT) which stretches from east of the Caledonian thrust front in central Jämtland to the Atlantic coast in western Tröndelag (Hurich et al., 1989; Palm et al., 1991; Hurich, 1996; Juhojuntti et al., 2001). The highly reflective upper crust shows a reflectivity pattern of crustal shortening consistent with surface observations, i.e. imbrication of allochthonous units and folding by major N–S to NE–SW-trending antiforms and synforms.

At the thrust front in central Sweden, Cambrian alum shales, deposited uncon-formably on the autochthonous crystalline basement, are separated from the overly-ing Caledonian allochthons by a major décollement (Gee et al., 1978). Comprehensive drilling programs targeting the metalliferous organic-rich alum shales (Gee et al., 1982) in the thrust front south of lake Storsjön reached about 30 km to the northwest, estab-lishing a 1–2° westwards dip of the décollement. At the Caledonian front in central Jämt-land, this major thrust, referred to here as the Jämtlandian décollement, coincides with the Caledonian sole thrust. We define the sole thrust to correspond to the lower limit of Caledonian deformation whereas the main décollement corresponds to the thrust zone that separates all the overlying allochthons from the less deformed basement. To the north of Storsjön, the inferred continuation westwards of this main décollement was traced along the CCT reflection seismic profile (Palm et al., 1991; Juhojuntti et al., 2001) to the Swedish-Norwegian border, where it appears to reach a depth of ca. 6 km (Hurich et al., 1989), beneath imbricated crystalline basement, in agreement with pre-vious modeling of refraction seismic (Palm, 1984), aeromagnetic (Dyrelius, 1980) and gravity data (Dyrelius, 1985; Elming, 1988). Magnetotelluric measurements along the Swedish section of the CCT profile (Korja et al., 2008), targeting the highly conductive alum shales, further support this interpretation.

A transition from thin-skinned (where deformation is mostly restricted to the al-lochthonous sediment-dominated units) to thick-skinned tectonics (with deep crustal

**SED**

doi:10.5194/se-2015-129

**Seismic imaging in the eastern Scandinavian Caledonides**

C. Juhlin et al.

deformation) is often attributed to large scale detachments and fault systems in the hinterland (Hurich, 1996; Mosar, 2003; Fossen et al., 2014) that are reactivated during post-collisional extension. In the case of the Caledonides, these are late-orogenic and involve NE–SW extension along the axis of the orogen. However, the previous thrust-
ing may well have been influenced by the pre-Caledonian geometry of the rifted and extended Neoproterozoic margin of Baltica (Gee et al., 2012).

Juhojuntti et al. (2001) identified a present day Moho at a depth of ca. 45–50 km beneath central Sweden and suggested deep crustal deformation in the subducting Baltica plate. However, the source of the strong reflections observed from within the
Palaeoproterozoic basement beneath Jämtland is yet to be determined. Two potential sources of the reflectivity patterns have been proposed (Palm et al., 1991; Juhojuntti et al., 2001), one being that they are related to the deformation history and the other that they are lithological in origin. Deformation zones could have developed during the Caledonian or Precambrian (Sveconorwegian, ca. 1.0 Ga, or older) orogenies. Al-
ternatively, most of the reflections could represent deformed mafic intrusions in the dominantly granitic basement rocks. Dolerite sills in the Siljan Ring area, a hundred kilometers to the southeast, are known to generate a similar seismic response (Juhlin, 1990). Dolerite sills and dykes are found to the south (0.95 Ga, Juhlin, 1990; Högdahl et al., 2004) and east (1.25 Ga, Högdahl et al., 2004; Söderlund et al., 2006) of the
thrust front of the central Scandinavian Caledonides and also in the Olden Window (Sjöström and Talbot, 1987).

The Swedish Scientific Drilling Program (SSDP) is operating within the framework of the International Continental Scientific Drilling Program (ICDP) to investigate funda-mental questions of global importance that are well defined in Scandinavia and require
drilling. One of the major projects led by SSDP is the Collisional Orogeny in the Scandi-navian Caledonides (COSC) project (Gee et al., 2010; Lorenz et al., 2011). This project aims to improve our understanding of collisional orogeny through scientific deep drilling of selected targets in the Swedish Caledonides.

**SED**

doi:10.5194/se-2015-129

**Seismic imaging in the eastern Scandinavian Caledonides**

C. Juhlin et al.

Title Page

Abstract | Introduction

Conclusions | References

Tables | Figures

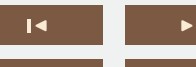

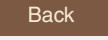 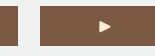

The first phase of the project, COSC-1, targeted the lower units of the high grade Seve Nappe Complex (SNC). These rocks originated along the rifted outer margin of continent Baltica, including the continent ocean transition (COT) zone (Andreasson, 1994), were partially subducted during the Ordovician and then emplaced hot onto
underlying allochthons. COSC-1 was drilled to a depth of 2.5 km with almost 100 % core recovery during May to August 2014 (Lorenz et al., 2015). The second phase, COSC-2, involves a second 2.5 km deep borehole that will start in the Lower Allochthon and aims to penetrate the Jämtlandian décollement as well as at least one of the underlying enigmatic basement reflectors. The focus of COSC-2 lies in understanding the thin-
skinned thrusting over the main décollement, the character of the deformation in the underlying crystalline Fennoscandian basement, and how this foreland deformation relates to the partial subduction of the Baltica margin in the hinterland (e.g. the Western Gneiss Region of southwestern Norway) in the early Devonian (Robinson et al., 2014).

In 2010, a 36 km long high resolution reflection seismic profile was acquired in the
Åre area (Fig. 1) with the purpose of finding the most suitable locations for the two scientific boreholes (Hedin et al., 2012). The location of the COSC-1 borehole was defined from these data (together with logistical considerations), but a location fulfilling the requirements of COSC-2 was not clearly identified. The interpreted main décollement and basement reflections appeared to continue shallowing towards the east and
the main seismic profile was therefore extended by about 17 km in 2011 and another ca. 14 km in 2014. A substantial gap in the 2011 acquisition was bridged in 2014 by an additional ca. 16 km long highly crooked profile south of the 2011 profile (Fig. 2).

Complementary to the seismic profiling, a magnetotelluric (MT) survey was conducted along the entire seismic profile in 2013 (Yan et al., 2016). Although this also
suffered from the need for a diversion, as with the seismic profile, it provided clear constraints on the depth to the top of the highly conductive alum shales. In addition, new aeromagnetic data were acquired by the Swedish Geological Survey in 2011, showing prominent features that may be linked with Rätan-type magnetite-rich granites in the basement.

Discussion Paper | Discussion Paper | Discussion Paper | Discussion Paper |

**SED**

doi:10.5194/se-2015-129

**Seismic imaging in the eastern Scandinavian Caledonides**

C. Juhlin et al.

Title Page

Abstract | Introduction

Conclusions | References

Tables | Figures

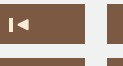 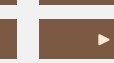

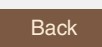 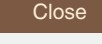

This paper focuses on the interpretation of the recently acquired seismic profiles, together referred to as the COSC seismic profiles (CSP), and the linking of these with the results from the drilling program in the late 1970's and observations from the COSC-1 borehole. In the light of the new geophysical data (reflection seismic, MT and aeromagnetic), we present an updated and extended interpretation of the seismic section from Hedin et al. (2012), along with alternative interpretations, with the main detachments being at both shallower and deeper levels than previously inferred. Based on our interpretations of the CSP data and the goals of the COSC scientific deep drilling project, we propose two candidate locations for the second borehole, COSC-2.

## 2  Caledonian geology and the central Jämtland profile

As mentioned above, the Caledonian allochthons in the thrust front of the orogen are separated from the underlying Precambrian crystalline basement by a major décollement. Along most of the orogenic front in Scandinavia and in the basement windows further west, this décollement is associated with Cambrian black alum shales (Andersson et al., 1985) which were deposited unconformably on the basement prior to thrust emplacement of the overlying nappes. These kerogen-rich shales, with carbon contents up to 15 %, acted as a lubricant to facilitate the low angle thrusting of the nappes for hundreds of kilometers onto the continental margin and platform of Baltica.

The Scandian nappes are commonly grouped into four major assemblages – Lower, Middle, Upper and Uppermost, as originally proposed for the Swedish Caledonides by Kulling (in Strand and Kulling, 1972), depending upon their level in the thrust system (Gee et al., 1985). Baltoscandian platform, inner margin and foreland basin strata dominate the Lower Allochthon. The outer margin and COT assemblages are generally thought to comprise the Middle Allochthon. Iapetus ocean-derived terranes characterize the Upper Allochthon and, at the top (Uppermost Allochthon), fragments of continental margin affinities are inferred to have been derived from Laurentia (Fig. 1). All these allochthons, together, are influenced by late orogenic shortening, with the de-

**SED**

doi:10.5194/se-2015-129

**Seismic imaging in the eastern Scandinavian Caledonides**

C. Juhlin et al.

velopment of major antiforms and synforms on N–S to NE–SW trending axes, many of the former exposing basement-cover relationships. In western Jämtland, the lithologies that comprise the Lower, Middle and Upper allochthons are well developed and distinct and occur with increasingly higher units present at the surface from east to west.

The Caledonian geology was mapped and compiled at 1 : 200 000 by Strömberg et al. (1984), and described by Karis and Strömberg (1998). Their work provides the basis for the map presented in Fig. 2. The bedrock geology of central and western Jämtland was summarized in the context of the COSC project by Gee et al. (2010). Therefore, we focus the geological overview in this paper on an ESE–WNW directed

profile that starts in the crystalline basement just east of Hackås (Fig. 2) and passes through the Jämtlandian Nappes, via Myrviken, where extensive drilling in the 1970's investigated the alum shales and the main décollement, as far west as Marby. A few kilometers farther west, near Hallen, the new seismic profiles (CSP) start and continue westwards through the Jämtlandian Nappes to merge into the 2010 profile that crosses

the Lower Seve Nappe and ends at Byxtjärn, just east of Åre (Fig. 2). The westernmost part of this profile, the Byxtjärn–Liten (BL) reflection seismic profile, was reported on in detail by Hedin et al. (2012).

Mapping of the many river sections transecting the Caledonian thrust front in the Scandes provided early investigators of the mountain belt with clear evidence of

a very gently W-dipping Precambrian basement surface (unconformity), overlain by thin autochthonous Cambrian sandstones and shales (locally also Neoproterozoic sandstones and tillites, and Ordovician limestones), beneath the main décollement. Prospecting for lead and zinc sulphide mineralizations in the sandstones (e.g. Grip, 1960; Saintilan et al., 2015), for example in the Laisvall and Vassbo areas (Fig. 1), pro-

vided supporting evidence for these observations. Subsequent, wide-ranging drilling programs by the Geological Survey of Sweden, targeting trace element concentrations in the metalliferous Cambrian Alum Shale Formation (Gee et al., 1982) and, more locally, in directly overlying limestones (Gee et al., 1978) defined the thrust front ge-

**[SED](doi:10.5194/se-2015-129)**

doi:10.5194/se-2015-129

**Seismic imaging in the eastern Scandinavian Caledonides**

C. Juhlin et al.



ometry to extend regularly westwards in the order of 30–40 km towards the hinterland, dipping at an angle of 1–2° to the west-northwest.

## 2.1 From the Caledonian front to Marby

In the Myrviken area in central Jämtland (Fig. 2), south of Storsjön, the drilling pro-
gram (Gee et al., 1982) defined the geometry of an exceptionally thick (up to 180 m) alum shale unit directly overlying the Caledonian sole thrust (here corresponding to the Jämtlandian décollement). Twenty-eight drillholes (all cored) provided the basis for identifying a major low grade uranium, vanadium, molybdenum, nickel resource in the organic-rich alum shales. Most of the holes also penetrated a thin sandstone-
dominated autochthonous Cambrian sedimentary succession overlying late Paleopro-
terozoic granites of the crystalline basement. Within the allochthonous units, both quartzites, stratigraphically underlying the alum shales, and limestones overlying them, occur in an imbricate stack that comprises the so-called Jämtlandian Nappes of the Lower Allochthon.

The above mentioned drillholes allow the décollement surface to be mapped in the Myrviken area (Fig. 2) and it shows the typical character of the Caledonian thrust front throughout most of the mountain belt. Interestingly, the fold axes in the allochthon in this area trend approximately N–S instead of NE–SW, possibly due to an anomalous basement high, ca. 50 km to the northeast in the Lockne area (Fig. 2), the result of a mid
Ordovician meteorite impact (Lindström et al., 1996). Cross-sections through the area of southern Storsjön illustrate the structure (Andersson et al., 1985) of the imbricate stack. Figure 3 shows a 25 km long profile trending NW, and partly NNW, from the thrust front near Hackås to Marby (Gee et al., 1982), oriented approximately parallel to the dip of the main décollement and sole thrust of the Jämtlandian Nappes. This
drillhole based profile ends about 10 km east of the easternmost end of the Dammån–Hallen (DH) seismic profile. If account is taken of the klippe (tectonic outlier) occurring to the south-southeast of Hackås in the Bingsta area, the main décollement can be

Discussion Paper | Discussion Paper | Discussion Paper | Discussion Paper | Discussion Paper |

**SED**

doi:10.5194/se-2015-129

**Seismic imaging in the eastern Scandinavian Caledonides**

C. Juhlin et al.

inferred to provide a regular surface, dipping about 1° west-northwest, over a distance of ca. 40 km.

## 2.2 From Hallen to Liten

The exposed and near surface bedrock between the village of Hallen and lake Liten
is dominated by Ordovician turbidites of the Jämtlandian Nappes. Only in the area of southeastern Liten are younger strata (lower Silurian, including quartzites and limestones) preserved locally in a shallow NW-trending syncline. The turbidites are folded on approximately N-trending axes and apparently imbricated by thrusting that is best exposed to the south in the N-plunging Oviksfjällen Antiform. The latter is inferred to
be a southern continuation of the Olden Antiform and, as shown on the Strömberg et al. (1984) map, comprises thrust sheets dominated by early Cambrian (perhaps late Ediacaran) quartzites, minor alum shales and subordinate slices of basement-derived felsic volcanic rocks, similar to the porphyritic rhyolites outcropping in the Mullfjället Antiform, to the west of the Åre Synform.

## 2.3 From Liten to Byxtjärn

Between Liten and Byxtjärn, near Undersåker, the seismic profile crosses the thrust between the Lower and Middle Allochthons. The former is composed of low to sub-greenschist facies Ordovician turbidites, locally passing up into early Silurian strata. In the hanging wall, the Seve Nappe Complex of the Middle Allochthon dips gently west-
wards in the eastern limb of the Åre Synform. It comprises mainly quartzites and subordinate calcsilicate-rich psammitic gneisses and marbles, with abundant amphibolitized dolerites and gabbros and some, usually isolated, ultramafites. These rocks comprise a highly reflective assemblage as found in the seismic investigations over the Åre and Tännfors synforms (Palm et al., 1991) and in the more recent seismic data presented
in Hedin et al. (2012). Along the thrust contact between the Seve Nappe Complex and the underlying strongly folded and intensely foliated turbidites of the Lower Allochthon

Discussion Paper | Discussion Paper | Discussion Paper | Discussion Paper | Discussion Paper |

**SED**

doi:10.5194/se-2015-129

**Seismic imaging in the eastern Scandinavian Caledonides**

C. Juhlin et al.

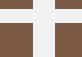

there occurs a sheet of felsic gneisses, locally underlain by a few tens of meters of ductilely deformed Särv Nappe metasandstones and concordant greenstones. Based on the seismic data acquired to date (Palm et al., 1991; Juhojuntti et al., 2001; Hedin et al., 2012), prominent reflective units that do not outcrop in the eastern limb of the
Åre Synform are expected to be present at depth.

Further west, in the western limb of the Åre Synform and the axial zone of the Mullfjället Antiform, Tiren (1981) mapped a detachment close above the basement and described relationships similar to those in the Caledonian front, i.e. with most of the quartzites, alum shales and overlying turbidites being allochthonous in relation to the
underlying Precambrian acid volcanic rocks with their thin veneer of alum shales and limestones.

## 3   Acquisition of the COSC seismic profiles (CSP)

Seismic acquisition parameters for the reflection seismic profiles from 2011 and 2014 were similar to those of the Byxtjärn–Liten (BL) and Kallsjön–Fröå (KF) segments, pre-
sented by Hedin et al. (2012) and summarized in Table 1. Crooked line acquisition was necessary along all the profiles due to the need to follow existing roads and paths. In general, an asymmetric split-spread geometry was employed that continuously moved with respect to the source. The acquisition varied slightly from profile to profile (depending on e.g. the terrain, road permissions, etc.). In addition, for the data acquired in
2014, changes were made to the source and recording equipment. The segments that make up the composite CSP, presented in this paper, are summarized below.

### 3.1   Byxtjärn–Liten (BL, 2010)

More than 1800 source points were activated along a 36 km long profile (Fig. 2) using a rock-breaking hydraulic hammer (VIBSIST) mounted on a front end loader. Nominal
source and receiver spacing was 20 m and a split spread of 360 active channels using

Discussion Paper | Discussion Paper | Discussion Paper | Discussion Paper |

**SED**

doi:10.5194/se-2015-129

**Seismic imaging in the eastern Scandinavian Caledonides**

C. Juhlin et al.

28 Hz geophones was rolled along with the source. In two locations of greater interest, the source point spacing was decreased to 10 m to increase the local fold. No source points were activated at the first 124 receiver locations (in the terrain) or along a few short parts in the western half (no permission to activate the source) resulting in a decreased fold in these areas. The fold along the entire profile therefore shows significant variation (Hedin et al., 2012).

## 3.2 Liten–Dammån (LD, 2011)

Acquisition of the Liten–Dammån profile used the same VIBSIST source as for the Byxtjärn–Liten profile. Permission to activate the source and plant receivers was not obtained along a nearly 4.5 km stretch of road close to the beginning of this profile. This created a gap in the acquisition geometry between the Byxtjärn–Liten and Liten–Dammån profiles (Fig. 2). This was partially bridged by using wireless receivers on the western side of the gap, coinciding with the last 1 km of the Byxtjärn–Liten profile, while wired receivers were placed on the eastern side. Source points were activated on both sides of the gap to undershoot it as much as possible. However, complete undershooting was not obtained.

## 3.3 Dammån–Hallen (DH, 2014)

The main profile of 2014 was the 14 km eastwards extension of the Byxtjärn–Liten and Liten–Dammån profiles, beginning at Dammån and ending south of Hallen (Fig. 2). Acquisition parameters for this profile differed from the Byxtjärn–Liten and Liten–Dammån profiles in that a different source and different geophones were used. Instead of 28 Hz geophones, 10 Hz geophones were used. More importantly, a lighter source was used. Instead of the VIBSIST source, a 400 kg weight-drop mounted on a small Bobcat excavator was used as a source. Previous studies (Sopher et al., 2014; Place et al., 2015) showed that this source could provide enough energy to image the subsurface to the

**SED**

doi:10.5194/se-2015-129

**Seismic imaging in the eastern Scandinavian Caledonides**

C. Juhlin et al.

depths of interest for the project, assuming thin Quaternary cover and shallow depths to bedrock.

### 3.4 Sällsjö (S, 2014)

To resolve the structures not imaged properly in the 4.5 km gap of the Liten–Dammån
profile, especially in the uppermost 2 km, a 16 km long profile was designed to fully
bridge this gap in 2014. Starting at the same location as the Liten–Dammån profile
and overlapping with the last 1 km of the Byxtjärn–Liten profile, the Sällsjö profile took
a more southern route via the village of Sällsjö before turning north and merging with
the Liten–Dammån profile (Fig. 2). Identical acquisition parameters to the Dammån–
Hallen profile were used, that is, the same source, recording system and spread.

### 4 Processing

Since drilling is targeted to 2.5 km and previous studies have shown source penetra-
tion depth generally to be to 5–6 km, only the first three seconds of data, correspond-
ing to ca. 9 km, were decoded and processed. Along the Byxtjärn–Liten and Liten–
Dammån profiles, where the VIBSIST data were acquired, decoding was performed
following Park (1996) and Cosma and Enescu (2001). 400–500 hits per source point
were stacked together to generate seismograms with a high $S/N$ ratio. For the data
acquired with the weight-drop source along the Sällsjö and Dammån–Hallen profiles,
the normally eight hits per source location were stacked together to similarly enhance
the $S/N$ ratio of the seismograms. The corresponding seismograms were then used as
input to a standard seismic processing package.

The vertical component data from the 3-component wireless receivers used in the
Liten–Dammån profile were extracted and merged with the 1-component receivers.
Noisy traces from bad source points (e.g. due to bad weather conditions, bad ground

Discussion Paper | Discussion Paper | Discussion Paper | Discussion Paper

**SED**

doi:10.5194/se-2015-129

**Seismic imaging in the eastern Scandinavian Caledonides**

C. Juhlin et al.

coupling) and receivers (e.g. due to bad ground coupling, instrument malfunction, environmental noise) were then removed prior to subsequent processing.

A smoothly curved crooked Common Midpoint (CMP) line was defined for the Byxtjärn–Liten and Dammån–Hallen profiles to minimize the number of missing traces while still following the acquisition line as closely as possible. Many of the structures in the area are sub-horizontal with a slight dip in the direction of acquisition. Therefore, it is possible (as shown below) to stack the midpoint traces of the Sällsjö profile, despite their far offset, together with those of the Liten–Dammån profile onto a straight CMP line segment between the Byxtjärn–Liten and Dammån–Hallen profiles and obtain a seismic section with coherent reflections.

In general, the processing followed a standard processing sequence (Table 2). However, as the VIBSIST and weight-drop data differed to some extent in their character due to the changed acquisition setups, pre-stack processing was performed separately for the different profiles. Examples of common source gathers from two locations along the profiles, before and after pre-stack processing, are shown in Fig. 4.

Thorough velocity analyses were performed in conjunction with both NMO and DMO corrections. DMO improved the coherency of the reflections along the Byxtjärn–Liten and Liten–Dammån profiles, but did not result in improved coherency along the Sällsjö and Dammån–Hallen profiles. The crookedness of the Sällsjö profile and the generally lower $S/N$ ratio along the Dammån–Hallen profile may explain the lack of improvement. Therefore, when the Liten–Dammån data were jointly processed with the Sällsjö data, as discussed below, no DMO was applied.

After processing the profiles separately, the Sällsjö and Liten–Dammån profiles were merged with the Dammån–Hallen profile to fill in the gap. Given that separate processing of the Sällsjö profile showed generally sub-horizontal reflections to be present below it, or reflections with NW dip (Fig. 5), the Sällsjö data and part of the Liten–Dammån data were projected onto a straight CDP processing line (Fig. 2). Likewise, the southeasterly part of the Liten–Dammån data were combined with the Dammån–Hallen data and processed along a straight CDP line (Fig. 2). Inspection of Fig. 5 shows that this

Discussion Paper | Discussion Paper | Discussion Paper | Discussion Paper |

**SED**

doi:10.5194/se-2015-129

**Seismic imaging in the eastern Scandinavian Caledonides**

C. Juhlin et al.

**SED**

doi:10.5194/se-2015-129

**Seismic imaging in the eastern Scandinavian Caledonides**

C. Juhlin et al.

methodology is generally justified even for the highly crooked Sällsjö profile. The general characteristics of the Liten–Dammån profile (Fig. 5a) are maintained in the merged section (Fig. 5c), while the projection of the data from the Sällsjö profile (Fig. 5b) fills in the gap due to the acquisition constraints. Although the details in the merged Liten–
Dammån and Sällsjö section may not be accurate, the general structure in the area is well represented.

Once processed, the separate profiles were projected onto a single profile to form a composite profile for interpretation and migration (Fig. 6). Since lateral variations in velocity are only minor, post stack time migration using a Stolt algorithm (Stolt, 1978)
was used. The decision to stack both the Liten–Dammån and Sällsjö profiles on the same straight CDP line parallel with the dip direction is also favorable for 2-D migration as this ensures that structures are moved to a more representative subsurface location. The migrated sections were finally time-to-depth converted to generate seismic sections suitable for geological interpretation. A velocity function based on the veloc-
ity analyses performed was smoothed to reduce the effects of local lateral variations, despite these being minimal, and used for the depth conversion. Figure 6b shows the section from Fig. 6a after migration and time-to-depth conversion.

## 5  Discussion

The interpretation of the Byxtjärn–Liten profile by Hedin et al. (2012) showed that the
high grade Seve Nappe Complex corresponds to a highly reflective unit, with a gently west-dipping eastern boundary in the vicinity of Undersåker (CDP 1200 in Fig. 6b), confirming previous evidence from the CCT profiling (Palm et al., 1991) in western Jämtland. Beneath and to the east of the Seve Nappe Complex, a transparent unit (ca. 1 km thick) is probably dominated by Ordovician turbidites, and underlain strati-
graphically by thin limestones and Cambrian alum shales. More flat-lying reflections were found below these exposed folded low grade metasediments of the Lower Allochthon. The main décollement was interpreted to be about 4.5 km below the Seve

Nappe Complex at Byxtjärn and to shallow eastwards to about 2.5 km in the vicinity of Liten, at CDP 3100. However, identification of the main décollement was uncertain due to lack of a continuous profile to the Caledonian front and ambiguities in the interpretation of the older CCT profile, where the uppermost crust is not so well imaged. The
new composite profile (CSP) presented here (Fig. 7) provides additional constraints on the structure, but a unique interpretation is still not possible. Below, we provide some general remarks on the CSP section and the relevance of other geophysical data for its interpretation. We then discuss interpretations of the seismic data, involving both a shallower and deeper main décollement than the one presented in the previous in-
terpretation of Hedin et al. (2012). Finally, we discuss two possible locations for the COSC-2 borehole.

### 5.1   General characteristics of the COSC composite seismic profile (CSP)

Both the VIBSIST source and weight-drop source generated enough energy to allow the seismic waves to generally penetrate to at least 9 km depth (Fig. 5). A direct com-
parison of the sources is not possible since the profile locations, acquisition geometries and the ambient noise conditions were not the same. In general, the VIBSIST source provided higher $S/N$ data than the weight drop source (compare Fig. 5a to b). However, merging of the two data sets generates a section which allows a clear correlation between reflections northwest and southeast of the gap on the Liten–Dammån pro-
file (Fig. 5c). In particular, after merging, it is clear that the sub-horizontal reflection at 0.7 seconds southeast of the gap (to the right of CDP 900 in Fig. 5c) is not connected to the two reflections at 0.4 to 0.6 seconds northwest of the gap (to the left of CDP 350 in Fig. 5c). Furthermore, the two west-dipping reflections at about 1 and 2 seconds (at CDP 1100, Fig. 5a), respectively, southeast of the gap appear to be connected to
the sub-horizontal reflections at 1.8 and 2.6 seconds northwest of the gap (Fig. 5a). Note that these reflections are better imaged on the Sällsjö profile with the weight-drop source than on the Liten–Dammån profile with the VIBSIST source (compare Fig. 5a with b at CDP 100 to 300).

Discussion Paper | Discussion Paper | Discussion Paper | Discussion Paper | Discussion Paper |

**SED**

doi:10.5194/se-2015-129

**Seismic imaging in the eastern Scandinavian Caledonides**

C. Juhlin et al.

**SED**

doi:10.5194/se-2015-129

**Seismic imaging in the eastern Scandinavian Caledonides**

C. Juhlin et al.

The entire composite profile (Fig. 7) shows generally sub-horizontal reflections in the uppermost 2 km. Below this depth the reflections are generally northwest dipping, but with some sub-horizontal reflections. An exception is the patchy highly reflective zone in the upper 2 km in the CSP interval from CDP 100 to CDP 1200, which charac-
terizes the Seve Nappe Complex. The west-dipping nature of this boundary is clearly defined from CDP 1100 to CDP 900, but the boundary becomes more diffuse below the central parts of the reflective zone. The diffuse nature of this boundary at depth was verified by the drilling of the COSC-1 borehole to 2.5 km (Lorenz et al., 2015) and the limited 3-D seismic survey that was acquired after drilling was completed (Hedin
et al., 2016). Between CDP 1100 and 4600 along the CSP, distinct, northwest-dipping reflections are present, some which can potentially be traced from 7 km depth to the sub-horizontal reflections between 1 and 2 km depth. These dipping reflections appear to sole into the overlying shallower sub-horizontal reflections. Similar dipping reflections were also observed on the CCT profile (Juhojuntti et al., 2001) and some of them can
be correlated to the CSP by their geometrical patterns in spite of the two profiles being separated by about 20 km. The source to these dipping reflections has previously been discussed (Palm et al., 1991; Juhojuntti et al., 2001; Hedin et al., 2012). Deformation zones, dolerite sheets, or a combination of the two were considered likely candidates. At the southeastern end of the CSP (CDP 4600 to 5500) the data quality deteriorates
significantly (Fig. 6) due to the presence of an up to 60 m thick sequence of unconsolidated Quaternary sediments which severely attenuate the signals and make it difficult to track the reflections beneath them. However, the shallowest sub-horizontal reflection can be traced to about 0.5 km depth at the southeasternmost end of the profile, as can the northwest-dipping reflection at about 6 km. The lack of clear reflections in between
these two is due to poor $S/N$. This reasoning is verified by comparing with the CCT profile (Fig. 8) on which there are clear northwest-dipping reflections in the equivalent depth interval at the same structural position along the profile. Note that the Dammån–Hallen profile was not extended further to the southeast due to permitting issues. Even

Discussion Paper | Discussion Paper | Discussion Paper | Discussion Paper |

if it had been possible, the thick sequence of unconsolidated sediments, also partly present to the southeast, would probably have made it difficult to acquire good data.

An important question is which reflection in the composite section represents the main Jämtlandian décollement, that is, the horizon which separates the more mobile overlying allochthons from the less deformed basement. The drilling in the Myrviken area clearly defined this surface southeast of the CSP. If the geometry of the main décollement in this area (Figs. 2 and 3) is projected into the southeastern end of the CSP, it would be expected to be found at a depth of about 500 m. This coincides with the sub-horizontal reflection found at this depth on the southeastern end of the composite section (Fig. 7). This reflection is not continuous northwestwards to CDP 4800, but rather irregular, probably due to the variable quality of the data that was acquired over the thick Quaternary sediments. However, we interpret the reflection at about 0.7 km depth at CDP 4800 along the CSP (Fig. 7) to represent the main décollement that was drilled further southeast in the Myrviken area. If so, this reflection can be fairly reliably traced along the CSP to CDP 3300. Here, it is unclear if the main décollement continues along the uppermost reflection at 1.2 km depth to CDP 2900 or along the lower one at 1.7 km depth at CDP 2900. Several lines of evidence indicate that the shallower reflection represents the Jämtlandian décollement. On the CCT profile, to the north, the main décollement was interpreted to be at about 1 km depth at the equivalent distance from the Caledonian front. Interpretation of the depth to magnetic basement based on the slope of the anomalies (using the standard Peter's method, see e.g. Reynolds, 2011) on the magnetic data along the composite profile (Fig. 7) gives depths of about 1.3 and 1 km at CDPs 3100 and 4100, respectively. Note that an alternative interpretation for the main décollement is that it deepens already at CDP 5200 down to 1 km depth. This alternative will be discussed later in the paper.

The new magnetotelluric (MT) survey along the profile (Yan et al., 2016) provides evidence of a gently undulating surface of the prominent uppermost conductive layer, shown in Fig. 7, this being located at ca. 500 m at the eastern end of the CSP, sinking to 600 m at CDP 3800, rising to 400 m at CDP 3500, sinking again to 1100 m at CDP 2700

Discussion Paper | Discussion Paper | Discussion Paper | Discussion Paper |

**SED**

doi:10.5194/se-2015-129

**Seismic imaging in the eastern Scandinavian Caledonides**

C. Juhlin et al.

and then rising again at CDP 1600 to 600 m, before dipping west again beneath the Seve Nappe Complex, west of Undersåker. This undulation fits well with the inferred location of the axes of the synforms and antiforms that are located in the vicinity to north and south of the CSP line. The highly conductive layer is interpreted to represent

the uppermost alum shales. It is therefore possible that the main décollement could be at a depth of about 1.5 km at CDP 2900 along the CSP and, if so, that it shallows to less than 1 km further west at CDP 1500 (Fig. 7) and then deepens at CDP 1300, below the Seve Nappe Complex.

Alternative interpretations accept the evidence for shallow décollements, but require

a substantially deeper location for the Caledonian sole thrust (e.g. Hedin et al., 2012). In both the Oviksfjällen and Olden antiforms, located to the south and north of the CSP profile, respectively, and apparently crossing it at ca. CDP 3300–3500, there is evidence of substantial shortening, with a quartzite-dominated thrust stack in Oviks-fjällen and much internal basement deformation in Olden. The Olden Antiform is of

particular interest because it contains an upper part of allochthonous basement (Gee, 1980; Robinson et al., 2014) thrust over the Cambro-Silurian sedimentary rocks of the Jämtland Supergroup. The extent to which sedimentary rocks of the Lower Allochthon might be represented at deeper structural levels than those exposed in the Olden and Oviksfjäll antiforms is impossible to say; MT methods have difficulty in detecting any

features below a strong conductor like the alum shales that is so well defined in the overlying décollement levels.

In the next section we focus on two alternative interpretations along the CSP. The first one is based on that in Hedin et al. (2012) with a deep main décollement; even deeper sole thrusts are also considered. The second one has a shallower main décollement,

more in line with the interpretations presented in Juhojuntti et al. (2001) and Korja et al. (2008). Note that even if the main décollement was interpreted as shallower in these studies the limit of Caledonian deformation was interpreted to be deep with a sole thrust reaching as deep as 15 km.

Discussion Paper | Discussion Paper | Discussion Paper | Discussion Paper |

**SED**

doi:10.5194/se-2015-129

**Seismic imaging in the eastern Scandinavian Caledonides**

C. Juhlin et al.

## 5.2 Interpretations

Figure 9 shows two possible interpretations of the composite profile, CSP. In Fig. 9a we present the section of Hedin et al. (2012) up to CDP 2900 (the easternmost extent of the Byxtjärn–Liten profile); further east we require a consistent prolongation within the CSP. Figure 9a shows the main décollement, here coinciding with the sole thrust, to continue gently upwards to the flat reflectors at about 2 km depth between CDP 3400 and 4200. It then ramps up to ca. 1.5 km and continues at this level to CDP 5100. Here it ramps up again to ca. 500 m and extends eastwards into the frontal décollement in the Myrviken area. The flat sections between CDP 3400 and 5100 both have hanging-wall west-dipping reflections, which suggests imbrication. The interpretation in Fig. 9a is based on the geology of the Oviksfjällen Antiform where the early Cambrian (to Edi-acaran) quartzites dominate, but include some slices of Precambrian volcanic rocks, particularly in the eastern limb of the structure. An even deeper Caledonian deforma-tion cannot be excluded. In this case, the sole thrust would extend from the frontal ramp at CDP 5100–5200 via a flat to CDP 4400, and from there downwards along prominent west-dipping reflections to a flat at ca. 5 km depth beneath CDP 3000 where it contin-ues westwards along more gently dipping reflections. This alternative would pass into the flat reflectors beneath the Mullfjället Antiform at ca. 7 km depth (Palm et al., 1991) and then perhaps extend beneath the Skardøra Antiform at similar (Hurich et al., 1989) or even greater depths (Gee, 1988). Both these "deep sole thrust" interpretations re-quire that the shallow Rätan-type basement beneath CDP 3400 and farther east, as suggested by the magnetic data, is allochthonous on top of a basement with similar characteristics.

An alternative interpretation is presented in Fig. 9b and includes new geophysical and geological evidence. The characteristic feature here is the shallow level of the main décollement to the southeast of CDP 1400, but with a significant deepening below the Åre Synform. The main décollement is accommodated within or in close proximity to the highly organic-rich Cambrian alum shales that constitute a weak horizon at the bot-

Discussion Paper | Discussion Paper | Discussion Paper | Discussion Paper | Discussion Paper |

**SED**

doi:10.5194/se-2015-129

**Seismic imaging in the eastern Scandinavian Caledonides**

C. Juhlin et al.

tom of the Early Paleozoic Baltoscandian sedimentary succession. Up to some tens of meters of Late Proterozoic Vemdalen quartzite separate the alum shales from the basement of the Fennoscandian Shield (Andersson et al., 1985). This unit may be locally absent because of the original basement topography or stripping by the overlying

thrust.

The new magnetotelluric (MT) data indicate the presence of a good conductor at ca. 1000 m at CDP 2200 and just below 500 m at CDP 4100 (Fig. 10). Below these depths, the reflection pattern in the seismic profile indicates imbricate thrusting above a detachment horizon. The latter is interpreted as the original, stratigraphic position of

the alum shales, which host the main Jämtlandian décollement. Within the imbricates, alum shale is brought to a shallower level as indicated by the MT data. This relationship is similar to that observed close to the present Caledonian front (Fig. 3; Gee et al., 1982; Andersson et al., 1985), where successions of alum shales with overlying Lower Ordovician limestones and shales and, occasionally, underlying Late Proterozoic Vem-

dalen quartzites are stacked to several times the original stratigraphic thickness.

Close to the northwestern end of the profile, in the Åre Synform, the 2.5 km deep COSC-1 drill hole provides control on the Lower Seve Nappe. At ca. 1700 m, the borehole enters a mylonite zone, representing a major thrust at the base of the Seve Nappe Complex. This zone extends to the bottom of the borehole, but a transition to rocks

of lower metamorphic grade, possibly from the Särv or Offerdal nappes, occurs at ca. 2350 m (Lorenz et al., 2015). Local 3-D reflection seismics at the drill site (Hedin et al., 2016) and a VSP survey in the drill hole (Krauß et al., 2015) suggest that the bottom of the thrust zone is located about 200 m below the total depth of the drill hole. The Särv and Offerdal Nappes are not continuous in the Åre area and seem to pinch

out towards the northeast somewhere below the Åre Synform, as indicated in Fig. 9b. However, farther east the Särv Nappe is present in klippen (Strömberg et al., 1984). It is also remarkable that, east of the Åre Synform, the fault zone that separates the Lower Seve Nappe from the Lower Allochthon is very narrow (north of CDP 1150), i.e. significantly different from the fault zone observed in the COSC-1 borehole. The

**SED**

doi:10.5194/se-2015-129

**Seismic imaging in the eastern Scandinavian Caledonides**

C. Juhlin et al.

fault zone observed at the surface is most likely a normal fault that places the Lower Seve Nappe against the Lower Allochthon and, thus, cuts out the tectonostratigraphy in-between. Similar relationships across faults were reported in the area west of Åre, both in Sweden and Norway (Sjöström et al., 1991; Braathen et al., 2000). Below 2 km

⁵ depth, the normal fault passes into a highly reflective zone above the interpreted main décollement, which it either cuts or merges into. The borders of the mylonite zone below the Åre Synform (dotted white lines in Fig. 9) trace along the reflectivity pattern eastwards towards location (1) in Fig. 9b, where also they merge into the above mentioned northwest-dipping highly reflective zone above the main décollement. East of

¹⁰ location (1), a prominent shallow basement reflection can be traced subhorizontally towards location (2), where it offsets the overlying reflections and continues upwards towards the southeast (broken line in Fig. 9b). It is interpreted as a thrust fault that at location (2) cuts upwards through the main décollement into the alum shale and brings basement with overlying rocks closer to the surface. The position, CDP 3100 and 3500,

¹⁵ corresponds well with the location of the Oviksfjällen Antiform, which about 10 km south of the seismic profile exposes Neoproterozoic bedrock in its core (Fig. 2). The nature of the reflectivity above this interpreted thrust fault is ambiguous. Possibly, it is similar to the basement reflections farther down. This would imply that the displacement along this particular reflector is a couple of kilometers, as indicated in Fig. 9b.

²⁰ For the section of the CSP, we suggest the following geological scenario: The high-grade metamorphic Lower Seve Nappe has a comparatively long tectonothermal history with migmatization as early as ca. 470 Ma (Li et al., 2014). Its initial emplacement as part of the Seve Nappe Complex has caused the penetrative ductile deformation with high internal strain, including gneisses with mylonitic fabric. Thrusting continuously

²⁵ progressed eastwards. Finally, the whole nappe stack with the Lower Allochthon at the bottom was translated towards the foreland on the main Jämtlandian décollement in the alum shales.

After metamorphic conditions in the Lower Seve Nappe had decreased considerably, the ca. 1 km thick mylonite zone began to develop by continued or resumed movement

Discussion Paper | Discussion Paper | Discussion Paper | Discussion Paper |

**SED**

doi:10.5194/se-2015-129

**Seismic imaging in the eastern Scandinavian Caledonides**

C. Juhlin et al.

**SED**

doi:10.5194/se-2015-129

**Seismic imaging in the eastern Scandinavian Caledonides**

C. Juhlin et al.

along the Seve–Särv boundary. The relative age of the movement on the normal fault that separates the Lower Seve Nappe from the Lower Allochthon east of the Åre Synform is not clear. However, Gee et al. (1994) suggested that Early Devonian extension on the Røragen detachment happened while thrusting was still going on. This could explain why both the reflective pattern that in the COSC-1 drill hole was related to the mylonite zone and the trace of the normal fault merge into a highly reflective zone that is directly overlying the main décollement at location (1) in Fig. 9b.

While nappe emplacement during Caledonian Orogeny progressed towards the foreland, Baltica was successively underthrusting Laurentia. Thus, it is very likely that also the Baltican basement experienced an eastwards progressing deformation, most likely above a sole thrust and possibly reactivating existing structures in the Proterozoic basement. Major orogen-parallel folding (e. g. Åre Synform, Tännforsen Antiform) occurred above this sole thrust. In the CSP, at least some of the deep reflections (around (3) in Fig. 9b) are thought to represent this basement deformation.

The development of the thick mylonite zone might be in part related to the development of the orogen parallel folds. Below the Åre Synform, folding of the main décollement could have translated the foreland directed movement upwards to straighten the deformation zone. At the basement culmination east of the Åre Synform, a similar process seems to have translated the deformation downwards into the basement along the probably pre-existing structure between CDP 1100 and 3100.

Scientific drilling at the COSC-2 site to 2.5 km will investigate and test the above scenario down into the shallow basement. It will sample at least one of the deep reflectors at its shallowest level and define its nature. Previously, these sub-horizontal to gently northwest dipping reflections have been interpreted as mafic sheets (Juhojuntti et al., 2001) hosted by Precambrian sandstones, volcanic rocks of Mullfjället type (Gee et al., 2010) or other igneous rocks that possibly are related to Trans-Scandinavian Igneous Belt (TIB) granites. This is similar to a setting observed in the autochthon south of the CSP where highly magnetic ca. 1700 Ma Rätan granites are associated with felsic volcanics and overlain by Precambrian sandstones with mafic volcanics.

Discussion Paper | Discussion Paper | Discussion Paper | Discussion Paper | Discussion Paper |

### 5.3 Locating the COSC-2 borehole

According to the COSC overall scientific targets, the COSC-2 borehole will investigate the metamorphic and structural evolution from the Lower Allochthon down into the basement of the Fennoscandian Shield. Important questions to be answered by the drilling are (i) is the metamorphic grade inverted in these middle to low grade green-schist facies rocks, (ii) were they also heated from above, (iii) what is the nature of the main décollement and where is it located and (iv) what structures generate the reflections in the Precambrian basement?

To reach these goals, the borehole will first drill the turbidites and limestones of the Lower Allochthon, penetrate the Cambrian alum shales and then continue downwards through thin (par)autochthonous Neoproterozoic to Ordovician sedimentary cover (quartzites, alum shales) and into the Precambrian crystalline basement, sampling a 1–1.5 km section of the latter.

Two possible locations for the COSC-2 borehole have been identified on the composite profile (Fig. 10). Option 1 is located along the Byxtjärn–Liten profile (Fig. 10a). Here, the two interpretations presented in the previous section differ from one another, with the main décollement being shallower in the new interpretation (Fig. 9b). Assuming that the main décollement has been correctly identified in Fig. 9b, the borehole will penetrate four reflectors in the autochthonous basement between about 1.4 and 2.2 km depth. A drill hole in this location would investigate the imbricate thrusting above the main décollement, whether the inferred deeper (shallow basement) thrust between CDP 1100 and 3100 is present, and, if not, what then causes the two shallower basement reflections. The two deeper basement reflections can be traced down to about 6 km northwest of the proposed site and appear to offset other reflections on the seismic section (Fig. 7). These two must surely originate in the Precambrian basement. One possible disadvantage with the location is that the separation between these four deeper reflections is small, at least on the present processing, and it may be difficult in the borehole to strictly identify the source to each of the four reflections. However,

Discussion Paper | Discussion Paper | Discussion Paper | Discussion Paper | Discussion Paper |

**SED**

doi:10.5194/se-2015-129

**Seismic imaging in the eastern Scandinavian Caledonides**

C. Juhlin et al.

Title Page

| Abstract | Introduction |
| Conclusions | References |
| Tables | Figures |

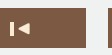 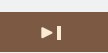

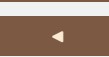 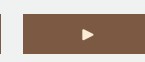

**SED**

doi:10.5194/se-2015-129

a combination of new high resolution seismic data and borehole seismic data should allow the source of the reflections to be determined without ambiguity. It is important to note that a Precambrian reflector may not be drilled if the Hedin et al. (2012) interpretation is correct since the main décollement will then be below the final depth of the
borehole. However, we regard this risk as minor, given the evidence from the new MT data and the constraints from the magnetic data.

Option 2 (Fig. 10b) is at a location (CDP 4100) where both interpretations presented in the previous section are similar, but with a large duplex structure below the alum shales between CDPs 3100 to 5200. The main décollement would be penetrated at
about 1700 m depth as defined by the seismic data, while the shallowest alum shale in the imbricate stacks above the main décollement would be at about 400 m as defined by the MT data. The main décollement at this location would then correspond to the strong sub-horizontal to gently northwest dipping reflection present across entire Fig. 10b between 1.9 and 1.3 km depth. The more steeply dipping shorter reflec-
tions above this strong reflection, but below the alum shales would represent boundaries between Cambrian strata and allochthonous Precambrian basement. At 2.2 to 2.3 km a basement reflector that appears to extend to depths of greater 7 km would be penetrated. The reflection from this structure is rather weak at the proposed site, but clearly present. A weaker dipping reflector, representing the boundary between
Cambrian meta-sedimentary rock and allochthonous basement would be penetrated at about 1.2 km.

## 6   Conclusions

The new seismic data acquired since 2011 in combination with the previous seismic data from 2010, the CCT profile, MT and magnetic data, and the drill holes in the
Myrviken area provide new constraints on the structure in this part of the Scandinavian Caledonides. The main décollement, as identified in the Myrviken drill holes, can be traced fairly confidently along the easternmost 20 km of the CSP, deepening along this

**SED**

doi:10.5194/se-2015-129

**Seismic imaging in the eastern Scandinavian Caledonides**

C. Juhlin et al.

section of the profile from about 0.5 km to nearly 1 km. Further west, in our preferred interpretation, the main décollement may continue to be relatively shallow, just somewhat greater than 1 km depth, and even shallowing on a structural high, before rapidly deepening just east of the Seve Nappe Complex in the eastern limb of the Åre Syn-
form. The previously acquired CCT profile, new MT data and magnetic data appear to be consistent with this interpretation. If correct, this requires the structural model presented in Hedin et al. (2012) to be revised. However, it is possible to interpret the main décollement to deepen already near the eastern end of the profile. If so, this deepening would be consistent with the Hedin et al. (2012) structural model. An even deeper level
to the west for the main décollement cannot be entirely ruled out.

Regardless of which interpretation is correct, the new data show mainly northwest dipping structures below the uppermost 1–2 km. Many of these structures have a similar pattern as those on the CCT profile located about 20 km to the north, suggesting large lateral continuity of the features out of the plane of the CSP. This is verified by
the highly crooked Sällsjö profile in which reflections can be traced more than 5 km to the south of the CSP. A definite interpretation of these northwest dipping reflections is not possible without drilling into them. The reflectivity pattern suggests that they are Caledonian or possibly reactivated older structures.

Two potential locations for the COSC-2 borehole have been identified along the CSP.
Drilling at the more westerly site would test which of the structural interpretations presented here is correct, the shallow main décollement or a deeper one. It would also penetrate four strong reflectors below the interpreted shallow main décollement. In the event that the Hedin et al. (2012) interpretation is correct then the main décollement would not be reached by the borehole. At the more easterly site the main décollement
would be penetrated at about 1700 m depth. Here, the main décollement is represented by a strong sub-horizontal reflection at about 1.7 km, an excellent drilling target, but its response is of an atypical nature compared to most of the other reflections. A more typical northwest dipping reflector is present below, but its reflectivity is somewhat diffuse at this potential site. Therefore, we favor the western site for the COSC-2 borehole.

*Acknowledgements.* The COSC project is a part of the Swedish Scientific Drilling Program (SSDP) which operates within the framework of the International Continental Scientific Drilling Program (ICDP) and the seismic reflection component of the project was funded by the Swedish Research Council (VR, grant 2013-5780). P. Hedin is also partly funded by VR. Hans Palm (HasSeis) planned and oversaw the seismic acquisition. GLOBE Claritas™ under license from the institute of Geological and Nuclear Sciences Limited, Lower Hutt, New Zealand was used to process the seismic data and seismic figures were prepared with GMT from P. Wessel and W. H. F. Smith. The applied geophysics group at Uppsala University is thanked for valuable discussions and advice throughout this work.

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

Title Page

| Abstract | Introduction |
| Conclusions | References |
| Tables | Figures |

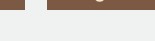

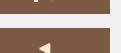 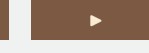

Back 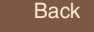 Close

Full Screen / Esc 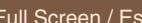

Interactive Discussion

Korja, T., Smirnov, M., Pedersen, L. B., and Gharibi, M.: Structure of the central Scandinavian Caledonides and the underlying Precambrian basement, new constraints from magnetotellurics, Geophys. J. Int., 175, 55–69, doi:10.1111/j.1365-246X.2008.03913.x, 2008.

Krauß, F., Simon, H., Giese, R., Buske, S., Hedin, P., and Juhlin, C.: Zero-offset VSP in the COSC-1 borehole, Geophysical Research Abstracts, 17, EGU2015-3255, 2015.

Labrousse, L., Hetényi, G., Raimbourg, H., Jolivet, L., and Andersen, T. B.: Initiation of crustal-scale thrusts triggered by metamorphic reactions at depth: insights from a comparison between the Himalayas and Scandinavian Caledonides, Tectonics, 29, 1–14, doi:10.1029/2009TC002602, 2010.

Ladenberger, A., Gee, D. G., Be'eri-Shlevin, Y., Claesson, S., and Majka, J.: The Scandian collision revisited – when did the orogeny start? Geophysical Research Abstracts, 14, EGU2015-12633, 2012.

Ladenberger, A., Be'eri-Shlevin, Y., Claesson, S., Gee, D. G., Majka, J., and Romanova, I. V.: Tectonometamorphic evolution of the Åreskutan Nappe – Caledonian history revealed by SIMS U-Pb zircon geochronology, in: New Perspectives on the Caledonides and Related Areas, edited by: Corfu, F., Gasser, D., and Chew, D. M., Geol. Soc. London, Spec. Publ., 390, 337–368, doi:10.1144/SP390.10, 2014.

Lindström, M., Sturkell, E., Törnberg, R., and Ormouml, J.: The marine impact crater at Lockne, central Sweden. GFF, 118, 193–206, doi:10.1080/11035899609546255, 1996.

Li, Y., Gee, D. G., Almqvist, B. S. G., Klonowska, I., Lorenz, H., Ladenberger, A., Majka, J., and Sjöström, H.: Mid Ordovician Leucogranites in the Lower Seve Nappe of central Jämtland, Swedish Caledonides, in: Abstract Volume Geological Society of Sweden, Lund, p. 118, 2014.

Lorenz, H., Gee, D., and Juhlin, C.: The Scandinavian Caledonides – Scientific Drilling at Mid-Crustal Level in a Palaeozoic Major Collisional Orogen, Sci. Dril., 11, 60–63, doi:10.5194/sd-11-60-2011, 2011.

Lorenz, H., Rosberg, J.-E., Juhlin, C., Bjelm, L., Almqvist, B. S. G., Berthet, T., Conze, R., Gee, D. G., Klonowska, I., Pascal, C., Pedersen, K., Roberts, N. M. W., and Tsang, C.-F.: COSC-1 – drilling of a subduction-related allochthon in the Palaeozoic Caledonide orogen of Scandinavia, Sci. Dril., 19, 1–11, doi:10.5194/sd-19-1-2015, 2015.

Majka, J., Be'eri-Shlevin, Y., Gee, D. G., Ladenberger, A., Claesson, S., Konečny, P., and Klonowska, I.: Multiple monazite growth in the Åreskutan migmatite: evidence for a poly-

**SED**

doi:10.5194/se-2015-129

**Seismic imaging in the eastern Scandinavian Caledonides**

C. Juhlin et al.

Discussion Paper | Discussion Paper | Discussion Paper | Discussion Paper | Discussion Paper

metamorphic Late Ordovician to Late Silurian evolution in the Seve Nappe Complex of west-central Jämtland, Sweden, J. Geosci., 57, 3–23, doi:10.3190/jgeosci.112, 2012.

Mosar, J.: Scandinavia's North Atlantic passive margin. J. Geophys. Res., 108, 2360, doi:10.1029/2002JB002134, 2003.

Palm, H.: Time-delay interpretation of seismic refraction data in the Caledonian front, Jämtland, central Scandinavian Caledonides, GFF, 106, 1–14, doi:10.1080/11035898409454597, 1984.

Palm, H., Gee, D. G., Dyrelius, D., and Björklund, L.: A reflection seismic image of Caledonian structure in Central Sweden, Geological Survey of Sweden, Ca 75, 1991.

Park, C. B.: Swept impact seismic technique (SIST), Geophysics, 61, 1789–1803, doi:10.1190/1.1444095, 1996.

Pascal, C., Ebbing, J., and Skilbrei, J. R.: Interplay between the Scandes and the Trans-Scandinavian Igneous Belt: integrated thermo-rheological and potential field modelling of the Central Scandes profile, Nor. Geol. Tidsskr., 87, 3–12, 2007.

Place, J., Malehmir, A., Högdahl, K., Juhlin, C., and Nilsson, K.: Seismic characterization of the Grängesberg iron deposit and its mining-induced structures, central Sweden, Interpretation, 3, SY41-SY56, doi:10.1190/INT-2014-0212.1, 2015.

Reynolds, J. M.: An Introduction to Applied and Environmental Geophysics, 2nd ed., Wiley-Blackwell, Chichester, UK, 2011.

Robinson, P., Roberts, D., Gee, D. G., and Solli, A.: A major synmetamorphic Early Devonian thrust and extensional fault system in the Mid Norway Caledonides: relevance to exhumation of HP and UHP rocks. in: New Perspectives on the Caledonides and Related Areas, edited by: Corfu, F., Gasser, D., Chew, D. M., Geol. Soc. London, Spec. Publ., 390, 241–270, doi:10.1144/SP390.24, 2014.

Saintilan, N. J., Stephens, M. B., Lundstam, E., and Fontboté, L.: Control of reactivated basement structures on sandstone-hosted Pb-Zn deposits along the Caledonian front, Sweden: evidence from airborne magnetic field data, structural analysis and ore grade modeling, Econ. Geol., 110, 91–117, doi:10.2113/econgeo.110.1.91, 2015.

Sjöström, H. and Talbot, C.: Caledonian and Post-Caledonian structure of the Olden Window,
Scandinavian Caledonides, GFF, 109, 359–361, doi:10.1080/11035898709453109, 1987.

Sjöström, H., Bergman, S., and Sokoutis, D.: Nappe geometry, basement structure and normal faulting in the central Scandinavian Caledonides; kinematic implications, GFF, 113, 265–269, doi:10.1080/11035899109453877, 1991.

Discussion Paper | Discussion Paper | Discussion Paper | Discussion Paper |

**SED**

doi:10.5194/se-2015-129

**Seismic imaging in the eastern Scandinavian Caledonides**

C. Juhlin et al.

Sopher, D., Juhlin, C., Huang, F., Ivandic, M., and Lueth, S.: Quantitative assessment of seismic source performance: feasibility of small and affordable seismic sources for long term monitoring at the Ketzin $CO_2$ storage site, Germany, J. Appl. Geophys., 107, 171–186, doi:10.1016/j.jappgeo.2014.05.016, 2014.

5  Stolt, R. H.: Migration by Fourier transform, Geophysics, 43, 23–48, doi:10.1190/1.1440826, 1978.

Strand, T. and Kulling, O.: Scandinavian Caledonides. Wiley Interscience, London, UK, New York, USA, 302 pp., 1972.

Strömberg, A., Karis, L., Zachrisson, E., Sjöstrand, T., Skoglund, R., Lundegårdh, P. H., Gor-
10  batschev, R., and Kornfält, K.-A.: Berggrundskarta över Jämtlands län utom förutvarande Fjällsjö kommun, scale 1 : 200 000, Geological Survey of Sweden, Ca 53 Geological Map, Uppsala, Sweden, 1984.

Söderlund, U., Elming, S.-Å., Ernst, R. E., and Schissel, D.: The Central Scandinavian Dolerite Group – Protracted hotspot activity or back-arc magmatism? Constraints from U-
15  Pb baddeleyite geochronology and Hf isotopic data, Precambrian Res., 150, 136–152, doi:10.1016/j.precamres.2006.07.004, 2006.

Tiren, S. A.: Thrust Nappe Geometry in the northern part of the Mullfjället Antiform, Jämtland Caledonides, Sweden, Terra Cognita, 1, 79, 1981.

Törnebohm, A. E.: Om fjällproblemet, GFF, 10, 328–336, doi:10.1080/11035898809444211,
20  1888.

Yan, P., Juanatey, M. A. G., Kalscheuer, T., Juhlin, C., Hedin, P., Savvaidis, A., and Lorenz, H.: A magnetotelluric investigation in the Scandinavian Caledonides in western Jämtland, central Sweden, using the COSC-1 borehole log as a priori information, Geophys. J. Int., in review, 2016.

Discussion Paper | Discussion Paper | Discussion Paper | Discussion Paper |

**SED**

doi:10.5194/se-2015-129

**Seismic imaging in the eastern Scandinavian Caledonides**

C. Juhlin et al.

**SED**

doi:10.5194/se-2015-129

**Seismic imaging in the eastern Scandinavian Caledonides**

C. Juhlin et al.

**Table 1.** Table 1. Acquisition parameters for the Byxtjärn–Liten profile (BL, 2010), Liten–Dammån profile (LD, 2011), Sällsjö profile (S, 2014) and Dammån–Hallen profile (DH, 2014).

| Profile | BL (2010) | LD (2011) | S (2014) | DH (2014) |
|---|---|---|---|---|
| Spread Type | Split spread | Split spread | Split spread | Split spread |
| Number of channels | 300–360 | 330–396 | 280–360 | 300–360 |
| Near offset | 0 m | 0 m | 0 m | 0 m |
| Maximum offset | 6804 m | 9502 m | 4633 m | 4634 m |
| Receiver spacing | 20 m | 20 m | 20 m | 20 m |
| Receiver type | 28 Hz, 1C | 28 Hz, 1C and 3C | 10 Hz, 1C | 10 Hz, 1C |
| Source spacing | 20 m (10 m) | 20 m | 20 m | 20 m |
| Source type | VIBSIST | VIBSIST | Weight drop | Weight drop |
| Hit interval for hammer | 100–400 ms | 100–400 ms | – | – |
| Sweeps per source point | 3–4 | 4–5 | – | – |
| Weight drops per source point | – | – | 8 | 8 |
| Nominal fold | 150–180 | 165–200 | 140–180 | 150–180 |
| Recording instrument | SERCEL 408 XL | SERCEL 428 XL | SERCEL 428 XL | SERCEL 428 XL |
| Field low cut | – | – | – | – |
| Field high cut | – | – | – | – |
| Sample rate | 1 ms | 1 ms | 1 ms | 1 ms |
| Record length | 26 s | 29 s | 28 s | 28 s |
| Profile length | ~ 36 km | ~ 17 km | ~ 16 km | ~ 14 km |
| Source points | 1807 | 638 | 767 | 626 |
| Data acquired | 30 Jul–13 Aug 2010 | 10–19 Oct 2011 | 18–24 Oct 2014 | 26–30 Oct 2014 |

Discussion Paper | Discussion Paper | Discussion Paper | Discussion Paper |

**SED**

doi:10.5194/se-2015-129

Seismic imaging in the eastern Scandinavian Caledonides

C. Juhlin et al.

**Table 2.** Processing steps and parameters for the Byxtjärn–Liten profile (BL, 2010), Liten–Dammån profile (LD, 2011), Sällsjö profile (S, 2014) and Dammån–Hallen profile (DH, 2014). BL, S+LD and DH were merged prior to migration to form the composite COSC Seismic Profile, CSP.

| BL (2010) | LD (2011) | S (2014) | S + LD | DH (2014) |
|---|---|---|---|---|
| Decoding of Vibsist data | Decoding of Vibsist data | Stacking of weight-drop gathers | | Stacking of weight-drop gathers |
| Manual Trace Edits | Manual Trace Edits | Manual Trace Edits | | Manual Trace Edits |
| Floating datum statics | Floating datum statics | Floating datum statics | | Floating datum statics |
| Refraction static corrections | Refraction static corrections | Refraction static corrections | Data merged | Refraction static corrections |
| Frontmute | Frontmute | | Surgical mute | Frontmute & Surgical mute |
| Spherical divergence compensation | Spherical divergence compensation | Spherical divergence compensation | Spherical divergence compensation | Spherical divergence compensation |
| Trace balancing | Trace balancing | Trace balancing | Trace balancing | Trace balancing |
| Wiener deconvolution | Wiener deconvolution | Spectral Equalization | Wiener deconvolution | Wiener deconvolution |
| | | Notch filter ($50 \pm 2$ Hz) | Notch filter ($50 \pm 2$ Hz) | Notch filter ($50 \pm 2$ Hz) |
| Band pass filter | Band pass filter | Band pass filter | Band pass filter | Band pass filter |
| 0–1 s, 25–50–80–120 Hz | 0–0.5 s, 25–50–100–150 Hz | 0–0.5 s, 25–50–100–150 Hz | 0–1 s, 25–50–100–150 Hz | 0–1 s, 25–50–100–150 Hz |
| 1.25–3 s, 20–40–80–120 Hz | 0.75–1.25 s, 20–40–90–135 Hz | 0.75–1.25 s, 20–40–90–135 Hz | 1.25–1.75 s, 20–40–90–135 Hz | 1.25–1.75 s, 20–40–90–135 Hz |
| | 1.75–3 s, 15–30–80–120 Hz | 1.75–3 s, 15–30–80–120 Hz | 2.25–3 s, 15–30–80–120 Hz | 2.25–3 s, 15–30–80–120 Hz |
| Airwave filter | Airwave filter | | | |
| Median velocity filter | Median velocity filter | Median velocity filter | Median velocity filter | Median velocity filter |
| 2200, 3200 m s$^{-1}$ | 2200, 3200 m s$^{-1}$ | 3100 m s$^{-1}$ | 1700, 3100 m s$^{-1}$ | 1700, 3100 m s$^{-1}$ |
| AGC (200 ms) | AGC (300 ms) | | AGC (200 ms) | AGC (500 ms) |
| Residual static corrections | Residual static corrections | | Residual static corrections | Residual static corrections |
| DMO & NMO correction | DMO & NMO correction | NMO correction | NMO correction | NMO correction |
| CMP stacking | CMP stacking | CMP stacking | CMP stacking | CMP stacking |
| Coherency filtering | Coherency filtering | Coherency filtering | Coherency filtering | Coherency filtering |
| (FX-Deconvolution) | (FX-Deconvolution) | (FX-Deconvolution) | (FX-Deconvolution) | (FX-Deconvolution) |
| | | Zeromute | Zeromute | Zeromute |
| FK-filter | FK-filter | | FK-filter | |
| Stolt migration | | | Stolt migration | Stolt migration |
| Time-to-Depth conversion | | | Time-to-Depth conversion | Time-to-Depth conversion |

Discussion Paper | Discussion Paper | Discussion Paper | Discussion Paper

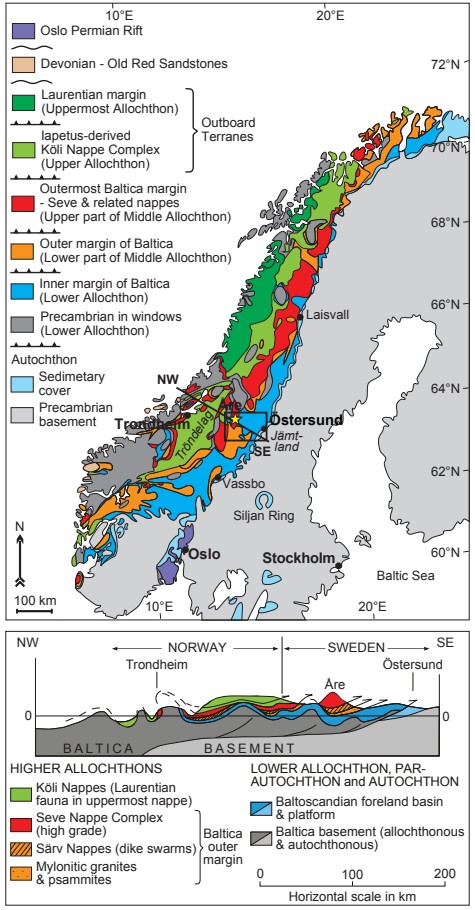

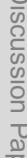

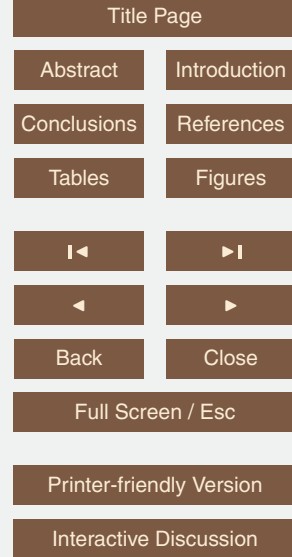

**SED**

doi:10.5194/se-2015-129

**Seismic imaging in the eastern Scandinavian Caledonides**

C. Juhlin et al.

**Figure 1. (a)** Tectonostratigraphic map of the Scandinavian Caledonides. The star marks the location of the COSC-1 borehole. **(b)** Schematic cross section along the NW–SE profile in **(a)** (modified from Gee et al.,1985).

**SED**

doi:10.5194/se-2015-129

**Seismic imaging in
the eastern
Scandinavian
Caledonides**

C. Juhlin et al.

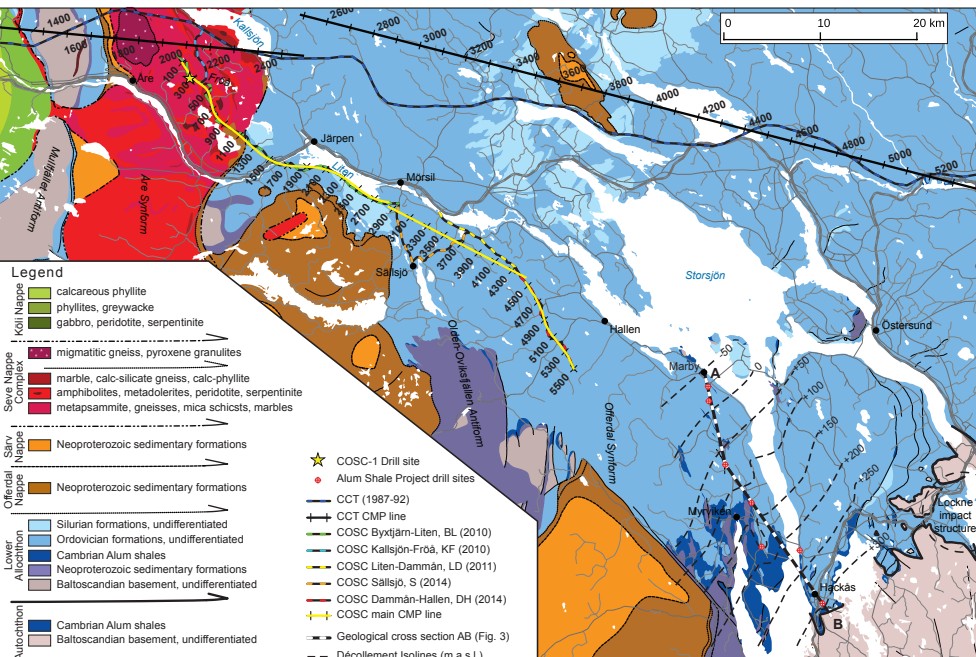

**Figure 2.** Regional bedrock geological map of western Jämtland showing the locations of the seismic profiles, the CCT seismic profile, the COSC-1 borehole and the shallow drillholes in the Myrviken area (based on the bedrock geological map of Sweden, [©]Geological Survey of Sweden [I2014/00601] and Strömberg et al., 1984). The location of the geological cross section shown in Fig. 3 is also indicated.

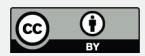

Discussion Paper | Discussion Paper | Discussion Paper | Discussion Paper | Discussion Paper |

**SED**

doi:10.5194/se-2015-129

**Seismic imaging in the eastern Scandinavian Caledonides**

C. Juhlin et al.

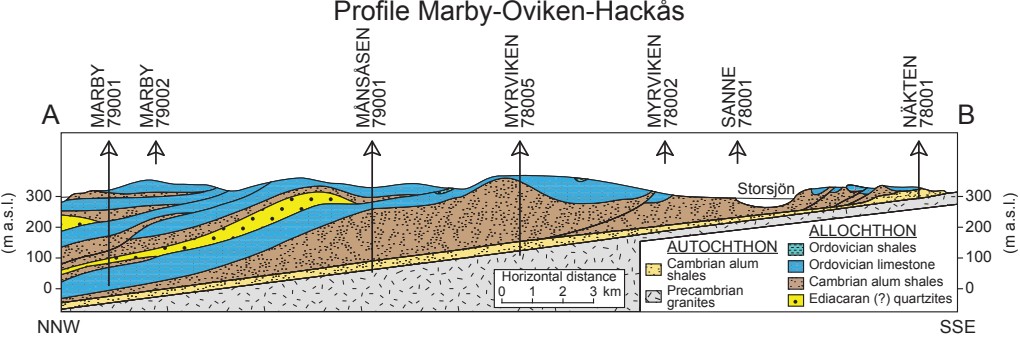

**Figure 3.** Geological cross section through the Myrviken area boreholes based on the SGU report on alum shales (Gee et al., 1982), shown at a vertical exaggeration of 10 : 1.

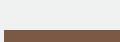

Byxtjärn-Liten profile source point 1596 (VIBSIST, 2010)

Sällsjö profile source point 9 (weight drop, 2014)

**Figure 4.** Two example source gathers before and after processing. **(a)** VIBSIST source gather from the Byxtjärn–Liten profile from the south shore of Lake Liten (Fig. 2) with only trace balancing applied. **(b)** The same source gather as in **(a)** after processing. **(c)** Weight-drop source gather from the Sällsjö profile from the eastern end of Lake Liten with only trace balancing applied. **(d)** The same source gather as in **(c)** after processing.

**SED**

doi:10.5194/se-2015-129

**Seismic imaging in the eastern Scandinavian Caledonides**

C. Juhlin et al.

Discussion Paper | Discussion Paper | Discussion Paper | Discussion Paper |

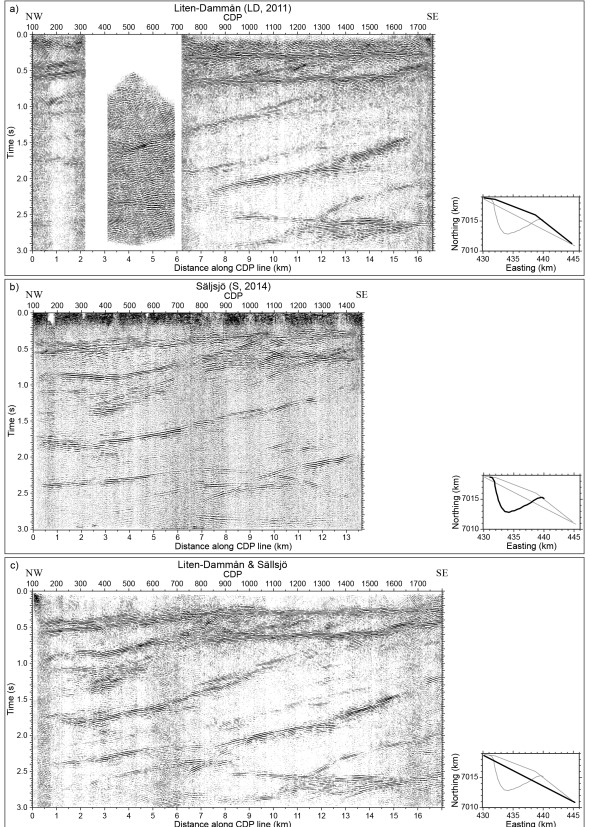

**Figure 5. (a)** Stacked section from the Liten–Dammån profile acquired in 2011 with the VIB-SIST source. **(b)** Stacked section from the Sällsjö profile acquired in 2014 with the weight-drop source. **(c)** Data from the Liten–Dammån and Sällsjö profiles processed together and stacked. The plan view maps show the three used CDP stacking lines with the thick black line indicating the CDP stacking line corresponding to the section shown in the same panel. **(a)** and **(c)** follow similar CDP stacking lines, while **(b)** follows a highly crooked CDP stacking line.

**SED**

doi:10.5194/se-2015-129

**Seismic imaging in the eastern Scandinavian Caledonides**

C. Juhlin et al.

Discussion Paper | Discussion Paper | Discussion Paper | Discussion Paper

**SED**

doi:10.5194/se-2015-129

**Seismic imaging in the eastern Scandinavian Caledonides**

C. Juhlin et al.

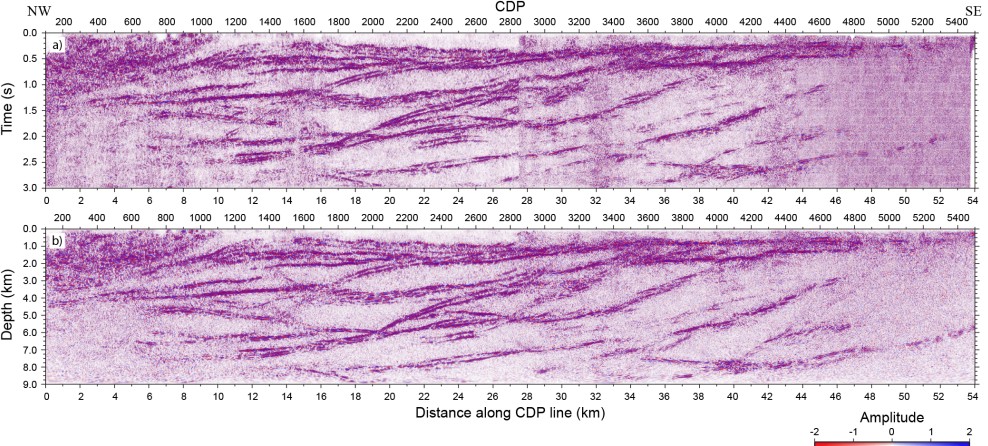
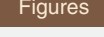
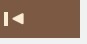
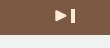
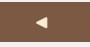
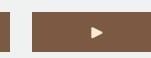
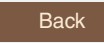
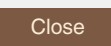
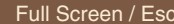

**Figure 6. (a)** Composite stacked section of the CSP. **(b)** Migrated and depth converted version of **(a)**. The CDP stacking line is shown in Fig. 2 with CDP numbers marked on the map. East of CDP 2850 the weight-drop source was employed.

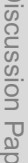

**SED**

doi:10.5194/se-2015-129

**Seismic imaging in the eastern Scandinavian Caledonides**

C. Juhlin et al.

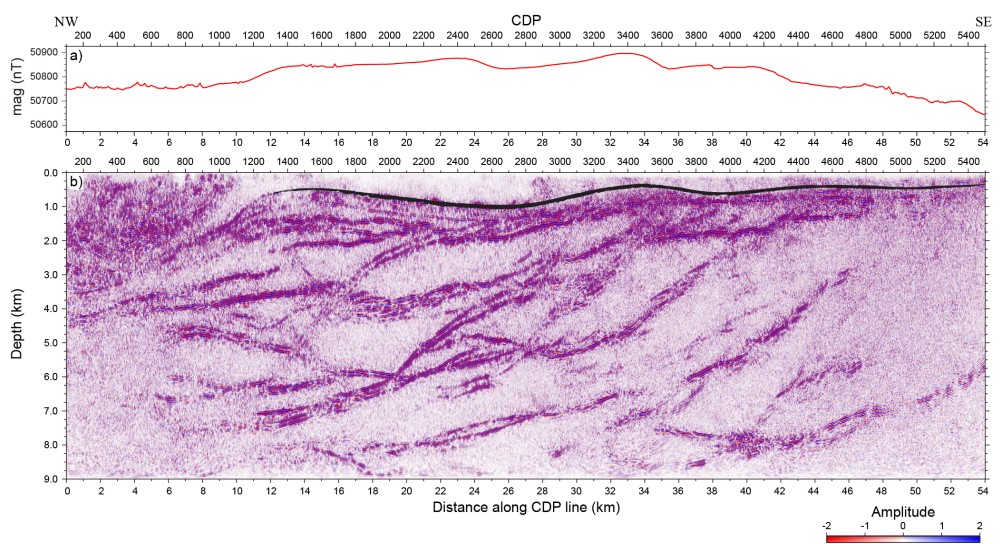

**Figure 7. (a)** Total magnetic field anomaly along the CSP. The anomalies at about CDP 1800, 3100 and 4100 can be interpreted as due to variations in the magnetic basement at depths of 1.3, 1.3 and 1.0 km, respectively. **(b)** Migrated and depth converted stack from Fig. 6 shown at a vertical exaggeration of 2 : 1. The black line marks the depth to the highly conductive layer from MT data as mapped by Yan et al. (2016). An excellent correspondence exists between the base of the uppermost seismically transparent zone and the mapped conductor. Therefore, the onset of reflectivity below the transparent zone is interpreted to represent the uppermost alum shale. Magnetic data are courtesy of the Geological Survey of Sweden (SGU).

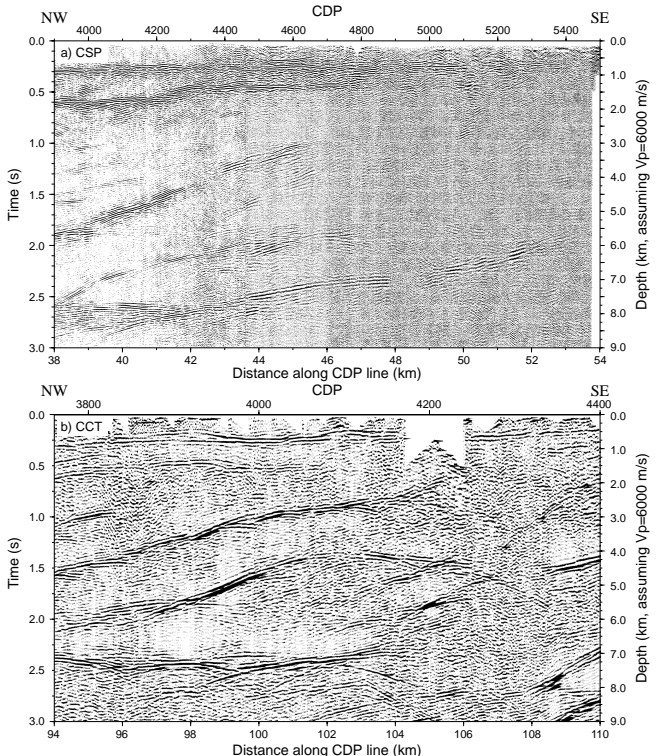

**Figure 8.** Sections of the CSP (top) and CCT profile (bottom) over approximately the same structural location. The three prominent reflective zones between 1 and 3 seconds on the western halves of the profiles are interpreted to represent the same structures. The transparent zone between 0.5 and 2 seconds on the eastern half of the CSP profile is interpreted as due to poor $S/N$ because of the thick sequence of loose sediments at the surface along this portion of the profile. Although data quality is variable at the equivalent location on the CCT profile, clear reflections are present between 0.5 and 2 seconds. It is likely that with better quality data, clear reflections would also be observed on the eastern half of the CSP between 0.5 and 2 seconds.

Discussion Paper | Discussion Paper | Discussion Paper | Discussion Paper |

**SED**

doi:10.5194/se-2015-129

**Seismic imaging in the eastern Scandinavian Caledonides**

C. Juhlin et al.

Discussion Paper | Discussion Paper | Discussion Paper | Discussion Paper |

**SED**

doi:10.5194/se-2015-129

**Seismic imaging in the eastern Scandinavian Caledonides**

C. Juhlin et al.

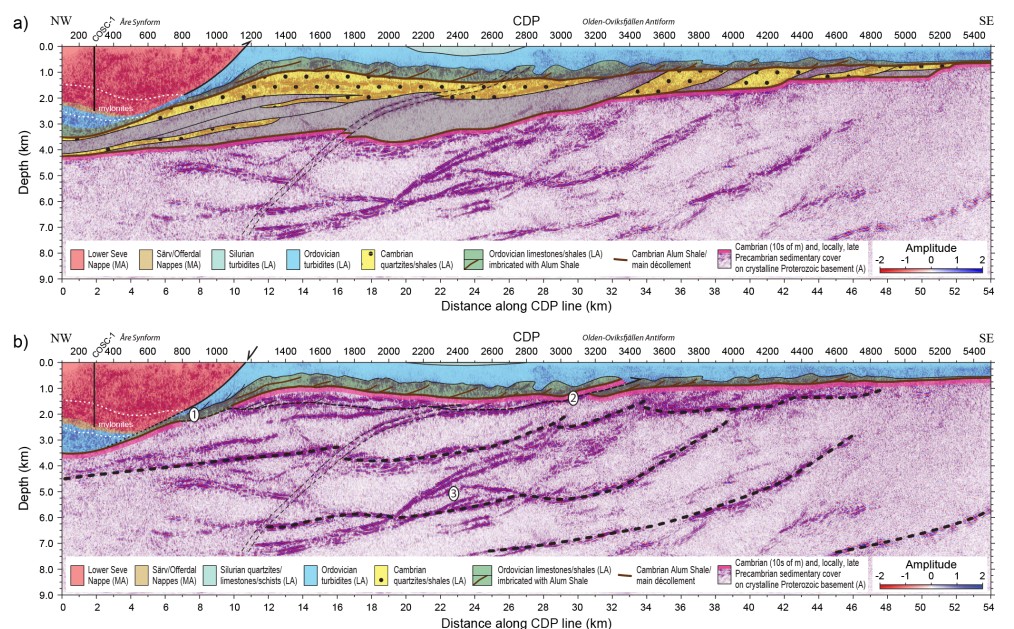

**Figure 9.** Two possible interpretations of the CSP data. In **(a)** the interpretation west of CDP 2800 is the same as in Hedin et al. (2012) with a deep main décollement and significant basement involved thrusting. In **(b)** the main décollement is much shallower in the west and lies only a few hundreds of meters below the top of the alum shales as interpreted from the CSP and the MT data. A second level of detachment might exist in the shallow basement reflectors below CDP 1000 to 3200. Numbers (1) and (2) are referenced in the text.

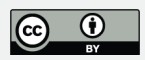

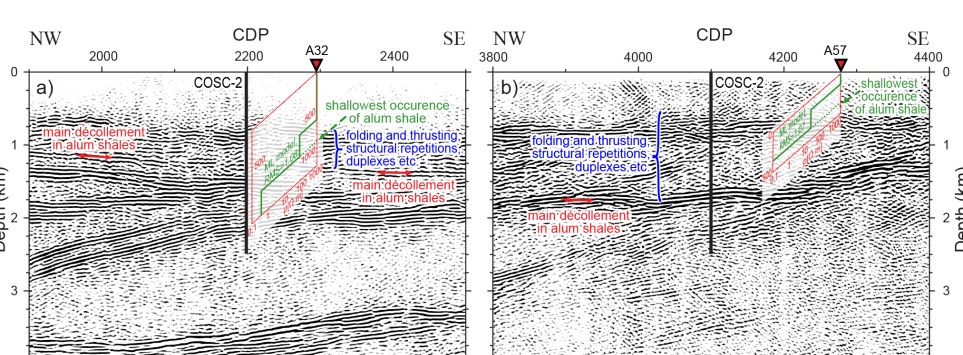

**Figure 10. (a)** Option 1 for the COSC-2 borehole corresponding to a location where the two interpretations in Fig. 9 differ significantly. Here, the main décollement would be penetrated at about 1.3 km depth if the interpretation in Fig. 9b is correct. Logistically, it is easier to place the borehole about 1 km to the east. Even at this location, two or three Precambrian reflectors would be penetrated. **(b)** In option 2 for the COSC-2 borehole the main décollement would be drilled at about 1700 m depth as interpreted in Fig. 9a. The rock between 800 m and 1.7 km is interpreted to consist of duplex structures. Conductivity profiles shown in the figures are placed at the locations of the MT stations that the inversions were preformed for. In **(a)** the uppermost alum shale would be penetrated at about 900 m depth and in **(b)** it would be penetrated at about 400 m depth.

**SED**

doi:10.5194/se-2015-129

**Seismic imaging in the eastern Scandinavian Caledonides**

C. Juhlin et al.

