# Peer review of "Seismic imaging in the eastern Scandinavian Caledonides: siting the 2.5 km deep COSC-2 borehole, central Sweden"

_Solid Earth, 2015_

## Short Comment (SC1) · 20 Jan 2016

We noted after acceptance of the discussion paper that the referencing of the magnetic anomalies to the CDP numbers is incorrect. A different version of the profile was used when the depth to the magnetic basement was being calculated. The correct locations for the paper are CDPs 1250, 2550 and 3550 with corresponding depths to magnetic basement of 1.3, 1.3 and 1.0 km, respectively. This correction concerns the caption to Figure 7a and Line 23 on page 18. Please take this correction into account when reading the paper.

---

## Referee Comment (RC1) · P. Ayarza (Referee) · 11 Feb 2016

**Comments on 'Seismic imaging in the eastern Scandinavian Caledonides: siting the 2.5 km deep COSC-2 borehole, central Sweden', by Juhlin, et al.**

This paper presents an excellent example of what we all should do to study continental collision and orogenic evolution. It is well written, easy to read and very interesting to follow. The models it proposes are excellent to open a discussion and key for the goal of the project: choosing the location of a 2.5 km depth borehole.

The focus of this work is a ca 55 km long, high resolution vertical incidence seismic profile acquired in Sweden as part of the COSC project, in an area that now represents the middle crust of the Caledonides. It samples part of what is called the Middle Allochtonous and the Lower Allochtonous. It is a complex area with outcropping Proterozoic and Paleozoic rocks affected by folding and thrusting. The basement has been probably affected by different deformation episodes, including Caledonian deformation. The authors present two alternative models to interpret their excellent composite seismic profile (CSP) in order to choose a location to carry out a 2.5 deep drill hole. These models are supported by geological, magnetic and magnetotelluric data in different degrees and both are coherent. Accordingly, it is very difficult to establish one of them as preferred model since both have strong supporting evidences.

**General comments**

To me, there are two key questions that could help to clarify the interpretations and that, either are not properly discussed in the text or I have missed the point. These two questions could help to discern the origin of the so called (and well deserved) enigmatic reflectivity and to decide which one of the two models is more viable.

1. Is the Jämtland detachment affected by the deformation/reflectivity observed underneath? This point deserves a bit more of discussion. If it is affected, then the reflectivity observed underneath has a high chance to the respond to a duplex/thrusts and this deformation is probably Caledonian, with a sole thrust running much deeper than the Jämtlandian detachment (model 9a). If it is not, then reflectivity (deformation) is previous and hasn't had reactivation during the Caledonian Orogeny. If it is slightly affected, the deformation might be previous to the Caledonian Orogeny but might have been slightly reactivated. At the sight of the profiles, the third case looks more likely and would support model 9b.

2. Is the Olden–Ovilksfjällen (O-O) antiform thrusted above the Jämtland detachment or it is underneath it. I find confusing evidences to discern this question in the map and the text. On the one hand, Figure 3 shows a cross-section where further to the SE, Neoproterozoic sedimentary formations seem to have been previously thrusted and the later development of the Jämtland detachment (JD) has carried them further east in a piggy-back style. If this also applies to the O-O antiform, it would be above the JD and would not need to be related with the reflectivity observed underneath it since it was probably folded before it was thrusted. In this case, model 9b would be more viable. It is true, however, that the reflectivity observed underneath the JD shows

some evidences of a basement high coinciding with the O-O antiform, but this deep reflectivity does not show any evidence of the Offerdel Synform suggesting that deformation above and below the JD might not be simultaneous and have no relation at all. In this case, morel 9b would be also more viable. On the other hand, the text seems to suggest that the O-O antiform is a basement culmination in which case, the JD is above it and should outcrop somewhere near this antiform. However, I don't see any evidences of that in the map. In this case, the 'enigmatic reflectivity' could of course be represented by model 9a and correspond to Caledonide basement imbrications. The sole thrust would run deeper than the JD and more coherence between deep reflections and surface features should be expected. Do the lithologies and the contacts (thrusts) observed in the O-O antiform support such a high reflectivity like the one observed underneath the JD?

Another question that could be explained in more detail is related to the high reflectivity observed in the Are synform and the Seve Nappe. Is this reflectivity similar to the one observed in CSP below the JD? When you say that prominent reflective units that do not outcrop in the eastern limb of the Are synform are expected at depth, do you mean in the middle allochthon or below? One possibility is that the 'enigmatic reflectivity' could be representing the extended outer margin of Baltica developed during the opening of the Iapetus(?). The reflectors could respond to normal faults, intruded by dykes and then slightly reactivated in the Caledonian orogeny. I guess this interpretation would also require estimations of shortening in the Seve Nappe and the imbricated Neoproterozoic outcrops. In any case, the pattern of the reflectivity is tectonic, i.e. that of a duplex (extensional and/or compressional). But the continuity of reflections suggests that lithology is also involved. In my opinion, faults are very heterogeneous and very seldom give such a continuous and well defined reflectivity unless they follow high impedance lithological boundaries. In this case, the magnetic highs could be related to these dykes. In fact, they appear more or less where these reflectors reach shallower levels (CDP's 2500, 3400, 4100 and even in 4800-5400, where even though the basement is supposed to be high, the magnetic field decreases but if has relative highs related to reflectors?).

**As very minor comments:**

Page 17: Line 11: …some OF which…?

Page 21: Line 26:…is present in A klippen?

Fig 7: Total magnetic field 'anomaly'? With values of 50000nT I think is not an anomaly but the total magnetic field. Also, is this data reduced to the pole?

In summary, I think this is a great paper that could be accepted as it is, although I would appreciate some more clarification/details regarding the comments posted above.

---

## Referee Comment (RC2) · D. White (Referee) · 14 Mar 2016

This paper presents beautiful seismic sections detailing the thrust stack west of the Caledonian thrust front. The interpretations are well supported by auxillary geological, drillhole and magnetotelluric data. The proposed drillhole locations each provide opportunities for confirming certain elements of the seismic interpretations and discriminating between the alternative interpretations that are presented. The demonstrated shallow inclination of the geology in the area validates the geometry of the reflectors as observed on the seismic sections and may allow drilling with some confidence in the absence of 3D seismic data from the proposed drill sites. However, acquisition of 3D data in advance of drilling would still be prudent. Furthermore, it will help in correlating

the reflections to drillhole geology particularly if the eventual drill hole is offset from the 2D seismic profiles.

---

## Author Comment (AC1) · 13 Apr 2016

We thank the reviewers for their comments on the manuscript. Concerning the comments from reviewer #1 we agree that there could be more discussion. However, the main focus of the paper has been to present new geophysical data and to justify the need to drill. We expect a number of followup papers after publication that deal with the more geological oriented questions raised by reviewer #1. In particular the idea of the basement reflectivity being related to the opening of the Iapetus ocean. We also appreciate the comment concerning the need for greater clarification of the two different major thrust zones (the Caledonian sole thrust and the Jämtlandian décollement). Ambiguity in a previous paper (Hedin et al 2012) has been removed from the presently

revised manuscript, which is based on substantially more geophysical data.

Please find the fully annotated version of the revised manuscript as a supplement to this comment.

Comment 1: We agree with reviewer 1 that the option for the deformation pattern seen below the Jämtlandian décollement is likely a combination of Caledonian and pre-Caledonian. We have added the following sentences beginning at line 14 on page 23 in the original MS.

"Additional evidence for some Caledonian deformation is found where reflections present below the interpreted Jämtlandian detachment appear to continue through it and offset the interpreted alum shales. Perhaps the best example of this is between CDPs 2600 and 2800 (Fig. 6) where the "double reflection" may offset the detachment and appears to have disturbed the overlying alum shales."

Comment 2: At present we cannot say whether the Olden-Oviksfjällen (O-O) antiform is thrusted above or below the Jämtlandian décollement (JD). Based on the seismic data and correlation with the CSP it is most likely thrust over the JD. The exposure on the antiform is rather poor, but quartzites have been identified along with more mafic rocks. The contrast between these could generate reflections. However, the pronounced reflectivity below the JD is interpreted to originate from magnetic basement (magnetite bearing granitoids). Therefore, the only way to investigate the nature of the basement reflectivity is to drill it. We have added the following sentences beginning at line 21 on page 19 in the original MS.

"Furthermore, the Oviksfjällen and Olden antiforms do not have a strong magnetic signature. The depth extent of the basement reflectivity is on the order of 10 km and presumably originates in magnetic basement, therefore, it is not clear how these antiforms can be linked to the origin of the basement reflections."

Comment Seve Nappe reflectivity: In the paper by Hedin et al. (2016, www.sciencedirect.com/science/article/pii/S0040195115006769) the Seve Nappe reflectivity is discussed in the detail. In this paper, the reflective nature of the nappe is attributed to the contrast between amphibolite and gneiss, the main lithologies drilled to 1700 m in the 2.5 km deep COSC-1 borehole. This reflective pattern is quite different from what we observe in the interpreted basement further east. Therefore, we cannot use the results from the COSC-1 borehole to directly interpret any of the distinct basement reflections we see east of CDP 1200. We have added the following sentences beginning at line 5 on page 11 in the original MS.

"Results from the 2.5 km deep COSC-1 borehole show that the reflectivity of the Seve Nappe Complex is due to the contrast between the high metamorphic grade gneisses and amphibolites (Hedin et al., 2016). Some of the reflections originating from below the bottom of the borehole, interpreted not to be part of the Seve Nappe Complex, can be traced towards the east, but do not extend to the surface."

Minor comments:

Page 17: Line 11: ...some OF which...?

Fixed

Page 21: Line 26:...is present in A klippen?

Fixed

Fig 7: Total magnetic field 'anomaly'? With values of 50000nT I think is not an anomaly but the total magnetic field. Also, is this data reduced to the pole?

Data were not reduced to the pole since this is generally not necessary at high latitudes. The map represents the total magnetic field with variations in it. We refer to these variations as anomalies.

Please also note the supplement to this comment:
http://www.solid-earth-discuss.net/se-2015-129/se-2015-129-AC1-supplement.pdf

[Figure]

**Supplement:**

[revised manuscript text omitted]

At the thrust front in central Sweden, Cambrian alum shales, deposited unconformably on the autochthonous crystalline basement, are separated from the overlying Caledonian allochthons by a major décollement (Gee et al., 1978). Comprehensive drilling programs

**Borttaget:** 400

**Borttaget:** several hundred million years of targeting the metalliferous organic-rich alum shales (Gee et al., 1982) in the thrust front south of lake Storsjön reached about 30 km to the northwest, establishing a 1-2° westwards dip of the décollement. At the Caledonian front in central Jämtland, this major detachment coincides with the Caledonian sole thrust (see profile in Fig. 1). We define the main décollement (in Jämtland – the Jämtlandian décollement, at the base of the Jämtlandian

Nappes) as the thrust zone that separates all the overlying long-transported allochthons from the underlying less deformed basement. The sole thrust corresponds to the lower limit of

Caledonian deformation, i.e. involving both the long-transported allochthons and the underlying crystalline basement in and below the antiformal windows. Along the CCT

reflection seismic profile, the sole thrust in the western part (Palm et al., 1991) was inferred to ramp up eastwards and pass into the Jämtlandian décollement, as defined in areas north of Storsjön (Juhojuntti et al., 2001). The sole thrust defined by Palm et al (1991) beneath the

Åre Synform and Mullfjället Antiform was inferred to continue westwards to the Swedish-

Norwegian border, where it appears to reach a depth of c. 7 km (Hurich et al., 1989); perhaps deeper (Gee, 1988, Hurich 1996), beneath the imbricated crystalline basement of the Skardöra Antiform. This interpretation is in agreement with previous 
[revised manuscript text omitted]

borehole show that the reflectivity of the Seve Nappe Complex is due to the contrast
between the high metamorphic grade gneisses and amphibolites (Hedin et al., 2016). Some
of the reflections originating from below the bottom of the borehole, interpreted not to be part
of the Seve Nappe Complex, can be traced towards the east, but do not extend to the
surface.
In the western limb of the Åre Synform and the axial zone of the Mullfjället Antiform, Tiren
(1981) mapped a detachment close above the basement and described relationships similar
to those in the Caledonian front, i.e. with most of the quartzites, alum shales and overlying
turbidites being allochthonous in relation to the underlying Precambrian acid volcanic rocks
with their thin unconformable veneer of alum shales and limestones.

**3 Acquisition of the COSC seismic profiles (CSP)**

[revised manuscript text omitted]

Two important relationships need to be defined – the depth and character of the

Jämtlandian décollement, and also the thickness and character of the underlying basement that has been influenced by Caledonian deformation. The drilling in the Myrviken area, southeast of the CSP, clearly defined the Jämtlandian décollement, where Cambro-

Ordovician sedimentary rocks are thrust over a thin autochthonous sedimentary cover and the underlying basement shows no evidence of Caledonian deformation. If the geometry of the Jämtlandian décollement in this area (Figs. 2 and 3) is projected into the southeastern end of the CSP, it would be expected to be found at a depth of about 500 m, coinciding with the sub-horizontal reflection found at this depth on the southeastern end of the composite section (Fig. 7). This reflection is not continuous northwestwards to CDP 4800, but rather irregular, perhaps due to the variable quality of the data that was acquired over the thick

Quaternary sediments. However, we interpret the reflection at about 0.7 km depth at

CDP 4800 along the CSP (Fig. 7) to represent the Jämtlandian décollement that was drilled further southeast in the Myrviken area. This reflection can be fairly reliably traced along the

CSP to CDP 3300. Here, it is unclear if the décollement continues along the uppermost reflection at 1.2 km depth to CDP 2900 or along the lower one at 1.7 km depth at CDP 2900.

Several lines of evidence indicate that the shallower reflection probably represents the

Jämtlandian décollement. On the CCT profile, to the north, the Jämtlandian décollement was interpreted to be at about 1 km depth at an equivalent distance from the Caledonian front.

Interpretation of the depth to magnetic basement based on the slope of the magnetic anomalies (using the standard Peter's method, see e.g. Reynolds, 2011) along the composite profile (Fig. 7) gives values of about 1.3 km and 1 km at CDPs 3100 and 4100, respectively. Note that an alternative interpretation for the Jämtlandian décollement is that it deepens already at CDP 5200 down to 1 km depth. This alternative will be discussed later in the paper.

The new magnetotelluric (MT) survey along the profile (Yan et al., 2016) provides evidence of a gently undulating surface of the prominent uppermost conductive layer, shown in Fig. 7, this being located at c. 500 m at the eastern end of the CSP, sinking to 600 m at CDP 3800, rising to 400 m at CDP 3500, sinking again to 1100 m at CDP 2700 and then rising again at CDP 1600 to 600 m, before dipping west again beneath the Seve Nappe Complex, west of Undersåker. This undulation fits well with the inferred location of the axes of the synforms and antiforms that are located in the vicinity to the north and south of the CSP line. The highly conductive layer is interpreted to represent the uppermost alum shales. It is therefore possible that the Jämtlandian décollement could be at a depth of about 1.5 km at CDP 2900 along the CSP and, if so, that it shallows to less than 1 km further west at CDP 1500 (Fig. 7) and then deepens at CDP 1300, below the Seve Nappe Complex.

All alternative interpretations accept the evidence for shallow décollements and require a substantially deeper location for the Caledonian sole thrust (e.g. Hedin et al., 2012). In both the Oviksfjällen and Olden antiforms, located to the south and north of the CSP profile, respectively, and apparently crossing it at c. CDP 3300-3500, there is evidence of substantial shortening, with a quartzite-dominated thrust stack in Oviksfjällen and much internal basement deformation in Olden. The Olden Antiform is of particular interest because it contains an upper part of allochthonous basement (Gee, 1980; Robinson et al., 2014) thrust over the Cambro-Silurian sedimentary rocks of the Jämtland Supergroup. The extent to which sedimentary rocks of the Lower Allochthon might be represented at deeper structural levels than those exposed in the Olden and Oviksfjällen antiforms is, at present, impossible to say; MT methods have difficulty in detecting any features below a strong conductor like the alum shales that is so well defined in the overlying décollement levels. Furthermore, the Oviksfjällen and Olden antiforms do not have a strong magnetic signature. The depth extent of the basement reflectivity is on the order of 10 km and presumably originates in magnetic basement, therefore, it is not clear how these antiforms can be linked to the origin of the basement reflections.

**5.2 Interpretations**

In the following section we discuss alternative interpretations along the CSP. The first one, based on Hedin et al. (2012), focuses on the sole thrust and considers even deeper structural levels for the Caledonian deformation. The second considers the Jämtlandian décollement in relation to the location of the uppermost alum shales and the underlying flatlying reflectors in the upper 2 km of the crust, in line with the interpretations presented in Juhojuntti et al. (2001) and Korja et al. (2008). Figure 9 illustrates these interpretations.

In Figure 9a we present the section of Hedin et al. (2012) up to CDP 2900 (the easternmost extent of the Byxtjärn-Liten profile); further east we define a consistent prolongation within the CSP. The sole thrust, in western parts at about 4 km depth, rises eastwards to the flat reflectors at about 2 km depth between CDP 3400 and 4200. It then ramps up to c. 1.5 km and continues at this level to CDP 5100. Here it ramps up again to c. 500 m and extends eastwards into the frontal décollement in the Myrviken area. The flat sections between CDP 3400 and 5100 both underlie hanging-wall west-dipping reflections, which suggests imbrication. The interpretation in Fig. 9a is based on the geology of the Oviksfjällen Antiform where the early Cambrian (to Ediacaran) quartzites dominate, but include some slices of Precambrian volcanic rocks, particularly in the eastern limb of the structure. Even deeper Caledonian deformation cannot be excluded. In this case, the sole thrust would extend from the frontal ramp at CDP 5100-5200 via a flat to CDP 4400, and from there downwards along prominent west-dipping reflections to a flat at c. 5 km depth beneath CDP 3000 where it continues westwards along more gently dipping reflections. This alternative would pass into the flat reflectors beneath the Mullfjället Antiform at c. 7 km depth (Palm et al., 1991) and then perhaps extend beneath the Skardöra Antiform at similar (Hurich et al., 1989) or even greater depths (Gee, 1988). Both these "deep sole thrust" interpretations require that the shallow Rätan-type basement beneath CDP 3400 and farther east, as suggested by the magnetic data, is allochthonous on top of a basement with similar characteristics.

The interpretation presented in Fig. 9b, concentrates on the Jämtlandian décollement beneath the Jämtlandian Nappes and the new geophysical and geological evidence relevant to the uppermost 2 km of the crust. The characteristic feature here is the shallow level of the décollement to the southeast of CDP 1400 and the significant deepening westwards below the Åre Synform. The Jämtlandian décollement is probably accommodated within, or in close proximity to the highly organic-rich Cambrian alum shales that constitute a weak horizon at, or near, the base of the Early Paleozoic Baltoscandian sedimentary succession. In some areas, up to some tens of meters of Ediacaran to lower Cambrian quartzites separate the alum shales from the underlying basement of the Fennoscandian Shield (Andersson et al., 1985). This unit may be locally absent because of the original basement topography, or stripping by the overlying thrust; alternatively, it may be repeated several times within the décollement zone.

The new magnetotelluric (MT) data indicate the presence of a good conductor at c. 1000 m at CDP 2200 and just below 500 m at CDP 4100 (Fig. 10). Below these depths, the reflection pattern in the seismic profile indicates imbricate thrusting above a detachment

**Borttaget:** Figure 9 shows two possible interpretations of the composite profile, CSP.

**Borttaget:** require

**Borttaget:** Figure 9a shows the main décollement, here coinciding with the

**Borttaget:** to continue gently upwards

**Borttaget:** have

**Borttaget:** An even

**Borttaget:** Skardøra

**Borttaget:** An alternative

**Borttaget:** is

**Borttaget:** main

**Borttaget:** includes

**Borttaget:** main

**Borttaget:** , but with a

**Borttaget:** main

**Borttaget:** bottom

**Borttaget:** Up

**Borttaget:** Late Proterozoic Vemdalen quartzite

**Borttaget:** main horizon. The latter is interpreted as the original, stratigraphic position of the alum shales, which host the Jämtlandian décollement. Within the imbricates, alum shale is brought to a shallower level as indicated by the MT data. This relationship is similar to that observed close to the present Caledonian front (Fig. 3; Gee et al., 1982; Andersson et al., 1985), where successions of alum shales with overlying Lower Ordovician limestones and shales and, further west, underlying quartzites are stacked to several times the original stratigraphic thickness.

**5.3 Relationships between the Jämtlandian décollement and mylonites in the Lower Seve Nappe**

Close to the northwestern end of the profile, in the Åre Synform, the 2.5 km deep COSC-1 drill hole provides control on the Lower Seve Nappe. At c. 1700 m, the borehole enters a mylonite zone, representing a major thrust at the base of the Seve Nappe Complex. This zone extends to the bottom of the borehole, but a transition to rocks of lower metamorphic grade, possibly from the Särv or Offerdal nappes, occurs at c. 2350 m (Lorenz et al., 2015). Local 3D reflection seismics at the drill site (Hedin et al., 2016) and a VSP survey in the drill hole (Krauß et al., 2015) suggest that the base of the thrust zone is located about 200 m below the bottom of the drill hole. The Särv and Offerdal Nappes are not continuous in the Åre area, but thin and pinch out towards the northeast somewhere below the Åre Synform, as indicated in Fig. 9b. However, farther east the Särv Nappe is present in a klippen (Strömberg et al. 1984). It is also remarkable that, east of the Åre Synform, the fault zone that separates the Lower Seve Nappe from the Lower Allochthon is very narrow (north of CDP 1150), i.e. significantly different from the mylonite zone observed in the COSC-1 borehole. The contact observed at the surface is most likely a W-dipping normal fault that places the Lower Seve Nappe against the Lower Allochthon and, thus, cuts out the tectonostratigraphy in-between. Similar relationships across faults were reported in the area west of Mullfjället in Sweden and Skardöra in Norway (Sjöström et al. 1991, Braathen et al. 2000). Below 2 km depth, the normal fault passes into a highly reflective zone above the interpreted Jämtlandian décollement, which it either cuts or merges into. The borders of the mylonite zone below the Åre Synform (dotted white lines in Fig. 9) trace along the reflectivity pattern eastwards towards location (1) in Fig. 9b, where also they merge into the above-mentioned NW-dipping highly reflective zone above the Jämtlandian décollement. East of location (1), a prominent shallow basement reflection can be traced subhorizontally towards location (2), where it offsets the overlying reflections and continues upwards towards the southeast (broken line in Fig. 9b). It is interpreted as a thrust fault that at location (2) cuts upwards through the Jämtlandian décollement into the alum shale and brings basement with overlying rocks closer to the surface. The position, CDP 3100 and 3500, corresponds well with the location of the Oviksfjällen Antiform, which about 10 km south of the seismic profile exposes Ediacaran-Cambrian quartzites, in its core (Fig. 2). The nature of the reflectivity above this interpreted thrust fault is ambiguous. Possibly, it is similar to the basement reflections farther down. This would imply that the displacement along this particular reflector is a couple of kilometers, as indicated in Fig. 9b.

For the section of the CSP, we suggest the following geological scenario: The high-grade metamorphic Lower Seve Nappe has a comparatively long tectonothermal history with metamorphism and pegmatite intrusion, as early as c. 470 Ma (Li et al. 2014). Its subsequent emplacement as part of the Seve Nappe Complex has caused the penetrative ductile deformation with high internal strain, including gneisses with mylonitic fabric. Thrusting continuously progressed eastwards with the whole nappe stack, including the underlying units of the Middle Allochthon and Lower Allochthon translated farther towards the foreland on the Jämtlandian décollement in the alum shales.

After metamorphic conditions in the Lower Seve Nappe had decreased considerably, the c. 1 km thick mylonite zone began to develop by continued or resumed movement along the Seve-Särv boundary. The age of the movement on the interpreted normal fault that separates the Lower Seve Nappe from the Lower Allochthon, east of the Åre Synform, is probably Early Devonian, as suggested by Gee et al. (1994) for the Röragen detachment where movement was inferred to have occurred while thrusting was still going on at depth beneath the Vigelen Antiform. This could explain why both the reflective pattern that in the COSC-1 drill hole was related to the Seve mylonite zone and the trace of the normal fault merge into a highly reflective zone that is directly overlying the Jämtlandian décollement at location (1) in Fig. 9b.

While nappe emplacement during Caledonian Orogeny progressed towards the foreland, Baltica was successively underthrusting Laurentia. Thus, it is very likely that also the Baltican basement experienced an eastwards progressing deformation, most likely above a sole thrust and possibly reactivating existing structures in the Proterozoic basement. Major orogen-parallel folding (e. g. Åre Synform and Mullfjället Antiform) occurred above this sole thrust. In the CSP, at least some of the deep reflections (around location 3 in Fig. 9b) are thought to represent this basement deformation.

Additional evidence for some Caledonian deformation is found where reflections present below the interpreted Jämtlandian detachment appear to continue through it and offset the interpreted alum shales. Perhaps the best example of this is between CDPs 2600 and 2800 (Fig. 6) where the "double reflection" may offset the detachment and appears to have disturbed the overlying alum shales.

Scientific drilling at the COSC-2 site to 2.5 km will investigate and test the above scenario
down into the shallow basement. It will sample at least one of the deep reflectors at its
shallowest level and define its nature.

*5.4    Locating the COSC-2 borehole*

According to the COSC overall scientific targets, the COSC-2 borehole will investigate the
metamorphic and structural evolution from the Lower Allochthon down into the basement of
the Fennoscandian Shield. Important questions to be answered by the drilling are (i) what is
the nature of the Jämtlandian décollement and where is it located, (ii) is the metamorphic
grade inverted in the middle to low grade greenschist facies sedimentary rocks, (iii) were
they heated from above, (iv) what lithologies and structures generate the reflections in the
Precambrian basement, and (v) what is the timing of deformation at these structural levels?
To reach these goals, the borehole should first drill the turbidites and limestones of the
Lower Allochthon, penetrate the uppermost Cambrian alum shales and then continue
downwards in the zone of high reflectivity, probably with repetition of thin Ediacaran to
Ordovician sedimentary cover (quartzites, alum shales and limestones) and then through the
Jämtlandian décollement into the Precambrian crystalline basement, sampling at least a
1 km section of the latter.
Two possible locations for the COSC-2 borehole have been identified on the composite
profile (Fig. 10). Option 1 is located along the Byxtjärn-Liten profile at CDP 2200 (Fig. 10a).
Assuming that the Jämtlandian décollement has been correctly identified in Fig. 9b, the
borehole will penetrate four reflectors in the underlying basement between about 1.5 and
2.2 km depth. A drill hole in this location would investigate the imbricate thrusting above the
Jämtlandian décollement, whether the inferred deeper (shallow basement) thrust between
CDP 1100 and 3100 is present, and, if not, what then causes the two shallower basement
reflections. The two deeper basement reflections can be traced down to about 6 km
northwest of the proposed site and appear to offset other reflections on the seismic section
(Fig. 7). These two must surely be located in the Precambrian basement. One possible
disadvantage with the location is that the separation between these four deeper reflections is
small, at least on the present processing, and it may be difficult in the borehole to strictly
identify the source to each of the four reflections. However, a combination of new high
resolution seismic data and borehole seismic data should allow the source of the reflections
to be determined without ambiguity.
Option 2 (Fig. 10b) is at a location (CDP 4100) where the sole thrust appears to be
converging upwards towards the Jämtlandian décollement. The drill hole would penetrate the
latter, as defined by a zone of flat-lying reflectivity between CDPs 3100 to 5200, at about 500
m depth; as in the Myrviken drill holes, it would be overlain by the shallowest alum shales,

**Borttaget:** Previously, these sub-horizontal to gently northwest dipping reflections have been interpreted as mafic sheets (Palm et al. 1991. Juhojuntti et al., 2001) hosted by Precambrian sandstones, volcanic rocks of Mullfjället type (Gee et al.., 2010),) or other igneous rocks that possibly are related to Trans-Scandinavian Igneous Belt (TIB) granites. This is similar to a setting observed in the autochthon, south of the CSP, where highly magnetic c. 1700 Ma Rätan granites are associated with felsic volcanics and overlain by Precambrian sandstones with mafic volcanics.

**Borttaget:** main

**Borttaget:** these

**Borttaget:** ii

**Borttaget:** also

**Borttaget:** iii) what is the nature of the main décollement and where is it located and (

**Borttaget:** will

**Borttaget:** through

**Borttaget:** (par)autochthonous Neoproterozoic

**Borttaget:** ) and

**Borttaget:** -1.5

**Borttaget:** Here, the two interpretations presented in the previous section differ from one another, with the main décollement being shallower in the new interpretation (Fig. 9b).

**Borttaget:** main

**Borttaget:** autochthonous

**Borttaget:** 4

**Borttaget:** main

**Borttaget:** originate

**Borttaget:** It is important to note that a Precambrian reflector may not be drilled if the Hedin et al. (2012) interpretation is correct since the main décollement will then be below the final depth of the borehole. However, we regard this risk as minor, given the evidence from the new MT data and the constraints from the magnetic data.

**Borttaget:** main

**Borttaget:** both interpretations presented in the previous section are similar, but with

**Borttaget:** large duplex structure below the alum shales

**Borttaget:** . The main décollement would be penetrated

**Borttaget:** 1700 m

**Borttaget:** defined

**Borttaget:** seismic data, while the

**Borttaget:** shale in the top of which occur at 400 m, based on the MT data. The Jämtlandian décollement is underlain by a duplex structure, about 1 km thick, characterized by more steeply dipping, shorter reflections representing boundaries between Cambrian strata (quartzites and perhaps subordinate alum shales) and fragments of allochthonous Precambrian basement. The basal thrust of the duplex is a well-defined strong sub-horizontal to gently NW-dipping reflection, present across entire Fig. 10b, between 1.3 and 1.9 km depth; this probably corresponds to the Caledonian sole thrust. At 2.2 to 2.3 km depth, a basement reflector that appears to extend westwards to depths of greater than 7 km would be penetrated by this hole. The reflection from this structure is rather weak at the proposed site, but clearly present.

**6 Conclusions**

An integrated interpretation of the geophysical and drill hole data (CSP, CCT, MT data, aeromagnetics) provides new constraints on the structure in the central part of the Scandinavian Caledonides. The Jämtlandian décollement, as identified in the Myrviken drill holes of the Caledonian thrust front, can be confidently traced westwards along the easternmost 20 km of the CSP, deepening in this section of the profile from about 0.5 km to nearly 1 km. Further west, in our preferred interpretation, the Jämtlandian décollement continues to be relatively shallow, just somewhat greater than 1 km deep, even shallowing on a structural high, before rapidly deepening just east of the Seve Nappe Complex, in the eastern limb of the Åre Synform. The previously acquired CCT profile, together with new MT and magnetic data, are consistent with this interpretation of the Jämtlandian décollement; nevertheless, even somewhat deeper levels are possible.

The extent of Caledonian deformation below the Jämtlandian décollement and influencing the underlying basement, is less easily defined and the location of the Caledonian sole thrust remains enigmatic. It may indeed coincide with the surface defined by Hedin et al. (2012) at c. 4.5 km depth beneath Åre, and then shallow eastwards, ramping up to converge with the Jämtlandian décollement near the end of the CSP and in the Myrviken area. However, deeper levels for the sole thrust beneath the CSP and farther to the west cannot be ruled out. The new data show mainly northwest dipping structures below the uppermost 1-2 km. Many of these structures have a similar pattern as those on the CCT profile located about 20 km to the north, suggesting large lateral continuity of the features out of the plane of the CSP. This is verified by the highly crooked Sällsjö profile in which reflections can be traced more than 5 km to the south of the CSP. A definite interpretation of these NW-dipping reflections is not possible without drilling into them. The reflectivity pattern suggests that they are Caledonian, or possibly reactivated older structures.

Two potential locations for the COSC-2 borehole have been identified along the CSP. Drilling at the more westerly site, on the south side of Lake Liten, will penetrate the full Silurian to Ediacaran stratigraphy and allow detailed analysis of the structure of the Jämtlandian décollement, defined by strong flat-lying reflections. It will also penetrate four strong reflections below the interpreted Jämtlandian décollement, allowing identification of the composition, structural characteristics and timing of deformation of these features. Drilling at the alternative site, about twenty kilometers farther southeast, will provide important evidence about the Jämtlandian décollement and possibly also the sole thrust. However, it may fail to provide unambiguous evidence about the character of the typical NW-dipping reflections in the basement, their reflectivity being somewhat diffuse at this potential site. Therefore, we favor the western site for the COSC-2 borehole.

**Acknowledgements**

The COSC project is a part of the Swedish Scientific Drilling Program (SSDP) which operates within the framework of the International Continental Scientific Drilling Program (ICDP) and the seismic reflection component of the project was funded by the Swedish Research Council (VR, grant 2013-5780). P. Hedin is also partly funded by VR. Hans Palm (HasSeis) planned and oversaw the seismic acquisition. GLOBE Claritas™ under license from the institute of Geological and Nuclear Sciences Limited, Lower Hutt, New Zealand was used to process the seismic data and seismic figures were prepared with GMT from P. Wessel and W. H. F. Smith. The applied geophysics group at Uppsala University is thanked for valuable discussions and advice throughout this work. We thank reviewers Puy Ayarza and Don White for constructive feedback on this manuscript.

Borttaget: would test which of
Borttaget: lake
Borttaget: main
Borttaget: structural interpretations presented here is correct, the shallow main
Borttaget: or a deeper one.
Borttaget: would
Borttaget: reflectors
Borttaget: shallow main
Borttaget: kilometres
Borttaget: . In the event that the Hedin et al. (2012) interpretation is correct then
Borttaget: main
Borttaget: would not be reached by the borehole. At the more easterly site
Borttaget: main décollement would be penetrated at about 1700 m depth. Here,
Borttaget: main décollement is represented by a strong sub-horizontal reflection at about 1.7 km, an excellent drilling target, but its response is of an atypical nature compared to most
Borttaget: other reflections. A more
Borttaget: northwest
Borttaget: reflector is present below, but its
Borttaget: is

[revised manuscript text omitted]
 Notch filter (50 ± 2 Hz) | Wiener deconvolution Notch filter (50 ± 2 Hz) | Wiener deconvolution Notch filter (50 ± 2 Hz) |
| Band pass filter 0-1 s     25-50-80-120 Hz 1.25-3 s   20-40-80-120 Hz | Band pass filter 0-0.5 s     25-50-100-150 Hz 0.75-1.25 s 20-40-90-135 Hz 1.75-3 s     15-30-80-120 Hz | Band pass filter 0-0.5 s     25-50-100-150 Hz 0.75-1.25 s 20-40-90-135 Hz 1.75-3 s     15-30-80-120 Hz | Band pass filter 0-1 s     25-50-100-150 Hz 1.25-1.75s  20-40-90-135 Hz 2.25-3 s     15-30-80-120 Hz | Band pass filter 0-1 s     25-50-100-150 Hz 1.25-1.75s  20-40-90-135 Hz 2.25-3 s     15-30-80-120 Hz |
| Airwave filter | Airwave filter | | | |
| Median velocity filter 2200, 3200 m s$^{-1}$ | Median velocity filter 2200, 3200 m s$^{-1}$ | Median velocity filter 3100 m s$^{-1}$ | Median velocity filter 1700, 3100 m s$^{-1}$ | Median velocity filter 1700, 3100 m s$^{-1}$ |
| AGC (200 ms) | AGC (300 ms) | | AGC (200 ms) | AGC (500 ms) |
| Residual static corrections | Residual static corrections | Residual static corrections | Residual static corrections | Residual static corrections |
| DMO & NMO correction | DMO & NMO correction | NMO correction | NMO correction | NMO correction |
| CMP stacking | CMP stacking | CMP stacking | CMP stacking | CMP stacking |
| Coherency filtering (FX-Deconvolution) | Coherency filtering (FX-Deconvolution) | Coherency filtering (FX-Deconvolution) | Coherency filtering (FX-Deconvolution) | Coherency filtering (FX-Deconvolution) |
| | | Zeromute | Zeromute | Zeromute |
| FK-filter | FK-filter | | FK-filter | |
| Stolt migration | | | Stolt migration | Stolt migration |
| Time-to-Depth conversion | | | Time-to-Depth conversion | Time-to-Depth conversion |

**Figure Captions**

Figure 1. (a) Provenance interpretation of the Tectonostratigraphic Map of the Scandinavian Caledonides, modified from Gee et al. (1985). The star marks the location of the COSC-1 borehole. (b) Schematic cross section (vertical exaggeration x10) along the NW-SE profile in (a), from Gee et al. (2010). The autochthonous basement (light grey) is separated from the Caledonian deformed basement (dark grey) by the Scandian sole thrust.

Figure 2. Bedrock geological map of western Jämtland, based on the bedrock geological map of Sweden, © Geological Survey of Sweden [I2014/00601] and Strömberg et al., (1984), showing the locations of the CSP and CCT seismic profiles, the COSC-1 borehole and the shallow drill holes in the Myrviken area. The location of the geological cross section, shown in Fig. 3, is also indicated.

Figure 3. Geological cross section through the Myrviken area boreholes based on the SGU report on alum shales (Gee et al., 1982), shown at a vertical exaggeration of 10:1.

Figure 4. Two examples of source gathers before and after processing. (a) VIBSIST source gather from the Byxtjärn-Liten profile from the south shore of Lake Liten (Fig. 2) with only trace balancing applied. (b) The same source gather as in (a) after processing. (c) Weight-drop source gather from the Sällsjö profile from the eastern end of Lake Liten with only trace balancing applied. (d) The same source gather as in (c) after processing.

Figure 5. (a) Stacked section from the Liten-Dammån profile acquired in 2011 with the VIBSIST source. (b) Stacked section from the Sällsjö profile acquired in 2014 with the weight-drop source. (c) Data from the Liten-Dammån and Sällsjö profiles processed together and stacked. The plan view maps show the three used CDP stacking lines with the thick black line indicating the CDP stacking line corresponding to the section shown in the same panel. (a) and (c) follow similar CDP stacking lines, while (b) follows a highly crooked CDP stacking line.

Figure 6. (a) Composite stacked section of the CSP. (b) Migrated and depth converted version of (a). The CDP stacking line is shown in Fig. 2 with CDP numbers marked on the map. East of CDP 2850 the weight-drop source was employed.

Figure 7. (a) Total magnetic field along the CSP. The anomalies at about CDP 1800, 3100 and 4100 can be interpreted as due to variations in the magnetic basement at depths of 1.3 km, 1.3 km and 1.0 km, respectively. (b) Migrated and depth converted stack from Fig. 6 shown at a vertical exaggeration of 2:1. The black line marks the depth to the highly conductive layer from MT data as mapped by Yan et al. (2016). An excellent correspondence exists between the base of the uppermost seismically transparent zone

**Borttaget:** map

**Borttaget:** ) (modified

**Borttaget:** .,1985

**Borttaget:** Regional bedrock

**Borttaget:** CCT seismic profile, the

**Borttaget:** (Based on the bedrock geological map of Sweden, © Geological Survey of Sweden [I2014/00601] and Strömberg et al., 1984).

**Borttaget:** example

**Borttaget:** anomaly and the mapped conductor. Therefore, the onset of reflectivity below the transparent zone is interpreted to represent the top of the uppermost alum shale. Magnetic data are courtesy of the Geological Survey of Sweden (SGU).

Figure 8. Sections of the CSP (top) and CCT profile (bottom) over approximately the same structural location. The three prominent reflective zones between 1 and 3 seconds on the western halves of the profiles are interpreted to represent the same structures. The transparent zone between 0.5 and 2 seconds on the eastern half of the CSP profile is interpreted as due to poor S/N because of the thick sequence of loose sediments at the surface along this portion of the profile. Although data quality is variable at the equivalent location on the CCT profile, clear reflections are present between 0.5 and 2 seconds. It is likely that with better quality data, clear reflections would also be observed on the eastern half of the CSP between 0.5 and 2 seconds.

Figure 9. Interpretations of the CSP data. In (a) the focus is on the sole thrust. The interpretation west of CDP 2800 is the same as in Hedin et al. (2012) and shows significant basement involved thrusting; farther east, the sole thrust is shown to ramp up to join the

Jämtlandian décollement near the thrust front. In (b) the Jämtlandian décollement is shown to dip very gently westwards, lying only a few hundreds of meters below the top of the alum shales, as interpreted from the CSP and the MT data. A second level of detachment may exist in the shallow basement reflectors below CDP 1000 to 3200. Numbers (1), (2) and (3)

are referenced in the text.

Figure 10. (a) Option 1 for the COSC-2 borehole. Here, the Jämtlandian décollement would be penetrated at about 1.3-1.5 km depth, if the interpretation in Fig. 9b is correct.

Logistically, it is easier to place the borehole about 1 km to the east. Even at this location, two or three Precambrian reflectors would be penetrated. (b) In option 2 for the COSC-2

borehole, the Jämtlandian décollement would be drilled at about 500 m depth. The structure beneath the Jämtlandian décollement, down to about 1600m, is dominated by a duplex, probably consisting of sedimentary formations and basement-derived imbricates. The basal thrust of the duplex is inferred to coincide with the Caledonian sole thrust. The conductivity profiles shown in the figures are placed at the locations of the MT stations that the inversions were performed for. In (a) the uppermost alum shale would be penetrated at about 900 m depth and in (b) it would be penetrated at about 400 m depth.

Borttaget: Two possible interpretations

Borttaget: with a deep main décollement

Borttaget: main

Borttaget: main

Borttaget: much shallower in the west and lies

Borttaget: might

Borttaget: and

Borttaget: corresponding to a location where the two interpretations in Fig. 9 differ significantly.

Borttaget: main

Borttaget: main

Borttaget: 1700

Borttaget: as interpreted in Fig. 9a.

Borttaget: main

Borttaget: rock between 800 m and 1.7 km

Borttaget: interpreted to consist of

Borttaget: structures. Conductivity

**Figure 1**

[Figure]

**Figure 2**

[Figure]

Figure 3

**Profile Marby-Oviken-Hackås**

[Figure]

A
MARBY 79001
MARBY 79002
MÅNSÅSEN 79001
MYRVIKEN 78005
MYRVIKEN 78002
SANNE 78001
NÄKTEN 78001
B

Storsjön

NNW

SSE

(m a.s.l.)

(m a.s.l.)

Horizontal distance

1  2  3 km

AUTOCHTHON
Cambrian alum shales
Precambrian granites

ALLOCHTHON
Ordovician shales
Ordovician limestone
Cambrian alum shales
Ediacaran (?) quartzites

Figure 4

[Figure]

Figure 5

[Figure]

**Figure 6**

[Figure]

**Figure 7**

Figure 8

[Figure]

**Figure 9**

[Figure]

**Figure 10**

[Figure]